# Interventions that enhance health services for parents and infants to improve child development and social and emotional well-being in high-income countries: a systematic review

Lisa Hurt,[1] Shantini Paranjothy,[1] Patricia Jane Lucas,[2] Debbie Watson,[2] Mala Mann,[3] Lucy J Griffiths,[4] Samuel Ginja,[5] Tapio Paljarvi,[1] Jo Williams,[6] Mark A Bellis,[7] Raghu Lingam[5]

<sup>1</sup>Division of Population Medicine, Cardiff University School of Medicine, Cardiff, UK
<sup>2</sup>School for Policy Studies, University of Bristol, Bristol, UK
<sup>3</sup>Specialist Unit for Review Evidence, Cardiff University, Cardiff, UK
<sup>4</sup>Population, Policy and Practice Programme, Institute of Child Health, London, UK
<sup>5</sup>Institute of Health and Society, Newcastle University, Newcastle, UK
<sup>6</sup>Bristol City Council, Bristol, UK
<sup>7</sup>Public Health Wales, Cardiff, UK

**Correspondence to**
Dr Lisa Hurt; hurtl@cardiff.ac.uk

## ABSTRACT

**Background** Experiences in the first 1000 days of life have a critical influence on child development and health. Health services that provide support for families need evidence about how best to improve their provision.

**Methods** We systematically reviewed the evidence for interventions in high-income countries to improve child development by enhancing health service contact with parents from the antenatal period to 24 months postpartum. We searched 15 databases and trial registers for studies published in any language between 01 January 1996 and 01 April 2016. We also searched 58 programme or organisation websites and the electronic table of contents of eight journals.

**Results** Primary outcomes were motor, cognitive and language development, and social-emotional well-being measured to 39 months of age (to allow the interventions time to produce demonstrable effects). Results were reported using narrative synthesis due to the variation in study populations, intervention design and outcome measurement. 22 of the 12 986 studies identified met eligibility criteria. Using Grading of Recommendations Assessment, Development and Evaluation (GRADE) working group criteria, the quality of evidence overall was moderate to low. There was limited evidence for intervention effectiveness: positive effects were seen in 1/6 studies for motor development, 4/11 for language development, 4/8 for cognitive development and 3/19 for social-emotional well-being. However, most studies showing positive effects were at high/unclear risk of bias, within-study effects were inconsistent and negative effects were also seen. Intervention content and intensity varied greatly, but this was not associated with effectiveness.

**Conclusions** There is insufficient evidence that interventions currently available to enhance health service contacts up to 24 months postpartum are effective for improving child development. There is an urgent need for robust evaluation of existing interventions and to develop and evaluate novel interventions to enhance the offer to all families.

**PROSPERO registration number** CRD42015015468.

### Strengths and limitations of this study

► To our knowledge, this is the first systematic review of interventions that enhance health services to improve child development outcomes including social and emotional well-being outcomes in the very early years.
► We used a broad systematic search of the extensive literature in this field and searched many sources in addition to database searches.
► We reviewed a larger number of primary studies than previous reviews of interventions in the early years. Our conclusion is consistent with these reviews.
► It was not possible to conduct a meta-analysis due to the variation in the types of interventions and methods used to measure outcomes.
► We do not report parental outcomes and cannot comment on whether parents benefited from these enhancements.

## INTRODUCTION

Experiences in the first 1000 days of life have a crucial influence on child development and health.[1] Appropriate early child development (including physical, social and emotional, language and cognitive domains) has consistently been shown to be associated with good health and educational outcomes in childhood and consequent health and employment outcomes in adulthood.[2–4] Adopting a life course approach, including early intervention, is essential,[5] and investment is therefore needed in effective prenatal and postnatal services to optimise child health, well-being and developmental resilience.[6]

The content of health services to promote maternal and child health, delivered during pregnancy and the early years through primary care and home visits, varies across

countries. A recent review suggested that the best services in Europe are 'characterized by personalised ongoing support during pregnancy, choice in birth arrangements, postnatal support and advice, and paid parental leave for mothers and fathers'.[7] In most high-income settings, early years services also work to a 'proportionate universalism' model where care is available to all, irrespective of need, with enhanced support offered to families at high risk of adverse outcomes.[1]

There is high-quality global evidence to support the effectiveness of many components of early years services including elements of antenatal care and centre-based preschool provision.[7][8] Interventions to promote child development by enhancing routine health services in the early years have also been developed. However, most have been targeted at and evaluated with high-risk families or children with an identified condition.[9–11] An unacceptably high proportion of children in both high and low-income settings do not achieve expected early learning goals before they start school,[12] and it has been argued that targeted approaches alone may not be sufficient.[13] Interventions to enhance contacts with all parents in existing services may be more effective in improving child development outcomes for several reasons. First, not all children who need support are identified by a targeted approach.[14] Targeting can lead to stigmatisation resulting in poor uptake or adherence.[15] Embedding interventions within an existing service, such as health visiting, which provides ongoing and consistent support for parents, may also improve the interaction between health professionals and parents and improve access to care at a crucial time in their child's development, leading to improvements in child development outcomes.[11] A review of interventions in low and middle-income settings noted that there was great diversity in both the scope and focus of research in this area and concluded that parents in such settings 'need to be supported in providing nurturing care and protection in order for young children to achieve their developmental potential'.[16] However, the effectiveness of such interventions to enhance existing multidisciplinary services in high-income settings is not known.

Previous reviews of early interventions in high-income settings fail to provide a full picture of interventions relevant to public health policy and practice because they do not provide a comprehensive examination of child development outcomes in the very early years (ie, the period during which the human brain develops most rapidly[17]). Neither does the evidence base to date include social and emotional well-being outcomes nor are these consistently defined and articulated. The objective of this systematic review is to fill these gaps, by examining the effect of interventions designed to enhance health service contacts with all parents and children in the very early years (defined as the antenatal period to 24 months postpartum) on child development and social and emotional well-being outcomes. Our research question was developed in partnership with local policy-makers and provides evidence for policy.[18]

## METHODS

### Protocol and registration

The protocol for this systematic review was registered in the International Prospective Register of Systematic Reviews (PROSPERO CRD42015015468) on 12 January 2015. This review is reported in accordance with Preferred Reporting Items for Systematic Reviews and Meta-Analyses guidelines.[19]

### Inclusion and exclusion criteria

We included randomised controlled trials (RCT; with individual or cluster randomisation) in any language that were published or unpublished. The interventions of interest were ones delivered within existing multidisciplinary healthcare services that are the cornerstone of early years programmes and are available to all. The interventions may be delivered by a range of staff within these services. We included studies from the 76 countries and territories classified by the World Bank in July 2014 as 'high-income economies'. Studies published in any language were eligible for inclusion.

To capture the effects of interventions delivered in the very early years, we included programmes that were delivered at any time from the antenatal period to 24 months postpartum. Given that some programmes continue beyond the child's second birthday, we specified that studies would be included if the mean age of the children at the start of the intervention was less than or equal to 24 months. To allow time for these interventions to produce demonstrable effects, we included studies that examined outcomes to 39 months of age (given that not all studies would manage to assess children on their third birthday exactly).

Studies that selected participants from the general population or included all individuals from a specific neighbourhood (eg, an area-based programme defined on the basis of postcode or zip code, known as 'geographically targeted' programmes in this review) were included. Studies were excluded if they selected participants based on individual risk factors (eg, an individually assessed income threshold for participating families or parental illness) or specific clinical subgroups (such as preterm babies or children with specific diagnoses).

### Interventions

We included interventions that were provided within the framework of the existing healthcare system. They could be designed to augment routine healthcare provision for all children in different ways, for example, by improving the skills or parental capacity of the parents or the family, improving the interaction between health professionals and parents, improving access to healthcare for the parents or the child or including elements designed to promote a specific area of child development. These included training modules designed to be delivered to parents with the intention of improving child development outcomes or any resources (such as printed materials, films, Apps) that health professionals or their

support workers could use in their work with parents. Interventions could be delivered at home, in group-based settings (eg, in general, obstetric or paediatric practice, in hospitals or community settings), through telemedicine or via a combination of different methods.

There is an argument that these different approaches should be separated into different systematic reviews (or indeed separated by outcome). We, like others,[10 16] chose to include these in a single review to avoid divisions that were arbitrary from a developmental or service delivery perspective and to avoid multiple overlapping, small and/or empty reviews in a field with limited literature.

### Outcomes
The outcomes were motor development (fine and gross), cognitive development, language development (receptive and expressive), social and emotional well-being and global child development. We included studies that used validated tools to measure these outcomes. Where unvalidated tools were used, we considered these to be secondary outcomes. Studies were included if they measured outcomes at any time between 3 months of age and 39 months postpartum (specifically, where the average age of the children at outcome measurement was 39 months or less).

### Search strategy
We searched for articles published in any language between 01 January 1996 and 01 April 2016 in the following databases: Cochrane Central Register of Controlled Trials, Medline, Embase, Cumulative Index to Nursing and Allied Health Literature, PsycINFO, Web of Science, Scopus, Applied Social Sciences Index and Abstracts, Literatura Latino Americana em Ciências da Saúde, Sociological Abstracts, Social Services Abstract and OpenGrey; and the following trial registers: ClinicalTrials.gov, UK Clinical Trials Gateway, UK Clinical Trials Gateway and WHO International Clinical Trials Registry Platform. Given our focus on enhancement of existing health services, we restricted to studies published within 20 years of our study inception since health service change has been substantial in the mid to late 20th century. We used a combination of medical subject headings and free text including terms for interventions to enhance health service contacts combined with terms relating to child development outcomes. Terms for the interventions included those that listed the professional delivering the programme (including 'health visitor', 'community nurse', 'nurse', 'midwife', 'general practitioner', 'early years educator', 'parent educator') and programme names that were already known to the review team. The Medline search strategy is shown in online supplementary web appendix A. We also searched websites of 58 relevant programmes and organisations and the electronic table of contents (eTOC) of eight key journals for relevant studies published within the last 2 years (see online supplementary web appendix B for

a full list). Reference lists of included and key papers were reviewed, and authors contacted for additional data where necessary.

### Study selection
All references identified by the searches were downloaded into Endnote and duplicates removed. Titles and abstracts were screened for inclusion independently by two of three reviewers (LH and LJG or SP). Full-text versions were obtained for the papers potentially meeting the inclusion criteria and were screened independently by two of three reviewers (LH and LJG or SP). Disagreements were resolved through discussion and in consultation with others in the review group.

### Data abstraction process
A data extraction form designed for the requirements of this review was used, which included details on the characteristics of the included studies, the interventions studied and assessment of risk of bias and Grading of Recommendations, Assessment, Development and Evaluations (GRADE) working group criteria. Multiple publications and reports from the same trial were linked and compared for completeness and contradictions. Data from each paper were extracted independently and in duplicate (completed by LH and LJG or SG or SP or TP).

### Analysis
Risk of bias was assessed following Cochrane guidelines.[20] Due to variation in (1) the populations studied, (2) the design of the interventions and (3) the wide range of outcome measures used (both in terms of the child development domains and/or the instruments used to assess the outcomes), it was not possible to conduct a meta-analysis and results were reported using narrative synthesis. We specified a priori that we would examine the results stratified by (1) risk of bias, (2) the intensity of the intervention, (3) the age of the child at which the intervention was delivered, (4) whether the programme was available to all or geographically targeted and (5) sociodemographic characteristics of the families in the trial. We selected these variables as we hypothesised that they would help to identify the characteristics of the interventions most likely to be effective (eg, if high-intensity interventions were more effective than low-intensity ones) or the populations in which they were most likely to be effective (eg, if programmes recruiting from defined neighbourhoods were more effective than those made available to all).

An assessment of the intensity of each intervention was conducted independently and in duplicate (completed by LH and LJG or SG or SP or TP) based on seven criteria: (1) total number of visits; (2) total duration of the programme; (3) total number of contact hours; (4) frequency of visits; (5) number of components; (6) whether components were delivered directly to parents

and/or children and (7) whether the components were delivered on a one-to-one basis or in a group session. Using these seven characteristics, we categorised the overall intensity for each intervention as 'low', 'moderate' or 'high'. Two review authors made this assessment using subjective determination (as used in reference 21) rather than a predefined algorithm or a scoring system to allow for the diversity and complex combinations of components to be reflected in the categorisation. Finally, the quality of the overall evidence for each outcome was assessed using GRADE criteria.[22]

### Public involvement

This work was conducted in collaboration with the Bristol Network for Early Years Health and Well being (www.bonee.org) and a range of stakeholders have been

involved in the design and conduct of this initiative. Parents were not involved in the design and conduct of the review, but we are discussing the results and interpretation with parents.

### RESULTS

Fifteen thousand two hundred and eighty records were identified in the database searches (figure 1). Searches of relevant programme and organisation websites and eTOC searches yielded 83 additional records. Once all searches were combined and duplicates removed, 12 986 records remained. After title and abstract screening, 12 644 records that were outside the scope of the review were excluded (the vast majority of these because their

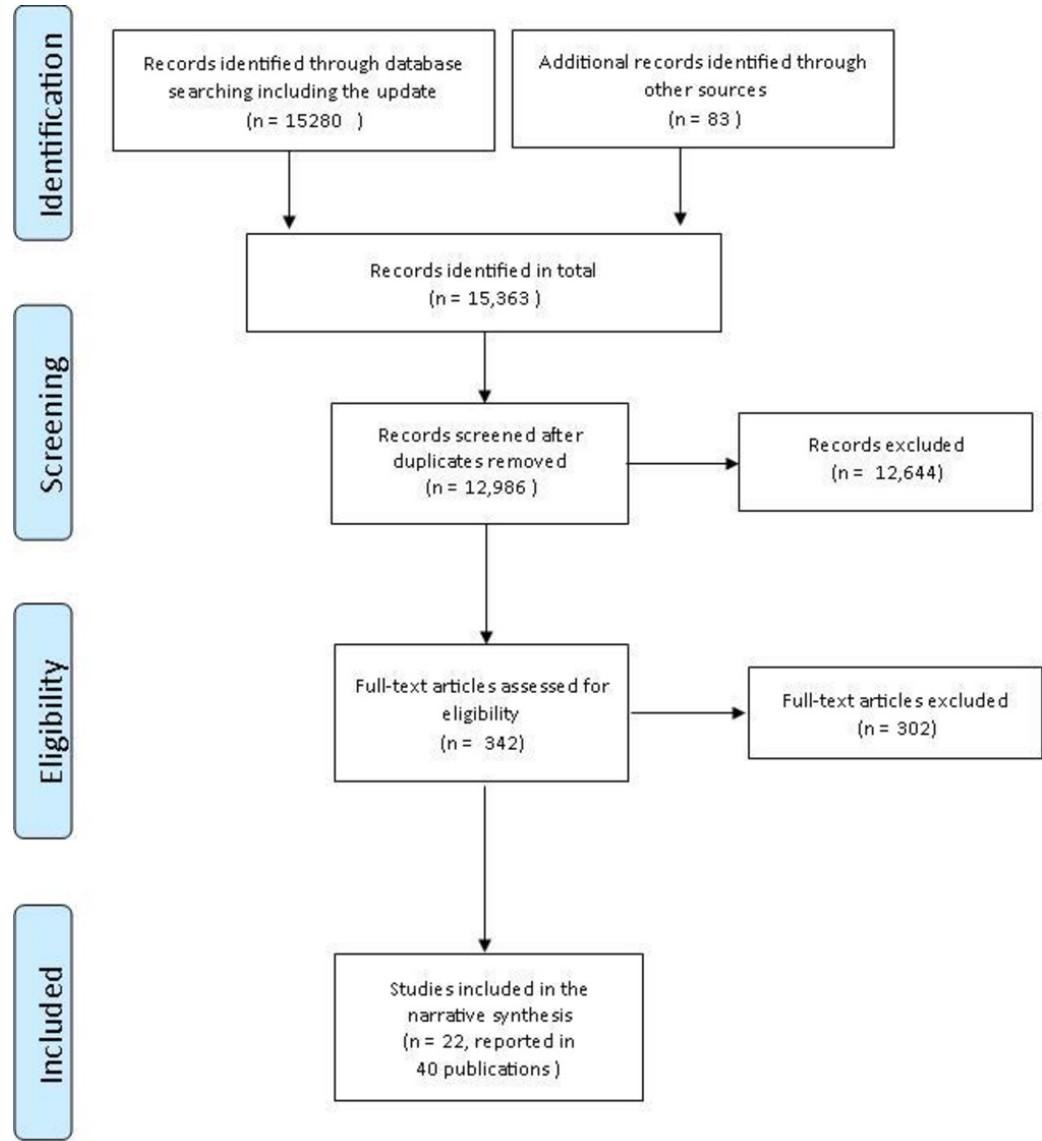

**Figure 1** Preferred Reporting Items for Systematic Reviews and Meta-Analyses flow diagram. Reason for exclusion at full-text screening: ongoing study, n=3; quasiexperimental (control group but no randomisation), n=10; pre–post test comparison only, n=5; not a primary study (reviews, editorials, programme descriptions), n=67; not conducted in a high-income country, n=3; intervention delivered in childcare settings, n=14; targeted programme (child factors), n=37; targeted programme (adult or family risk factors), n=88; mean age of children at intervention >24 months, n=53; mean age of children at outcome >36 months, n=5; no child development outcomes, n=17.

intervention was targeted at families at high-risk of adverse outcomes or at children with identified conditions). Of the 342 records included in full-text review, there were 22 RCTs that fulfilled our inclusion criteria (reported in 40 publications[23–62]). We also identified three relevant ongoing trials.[63–65] Reasons for exclusion are provided in figure 1.

## Trial characteristics

The 22 included trials are described in table 1. Three were cluster randomised (clinics[25 46] or healthcare workers[58]), with the remainder randomising individual mothers, parents or mother–child dyads. Ten were conducted in the USA, three in the UK, two in Canada, two in Ireland, one in Australia, one in Chile, one in Japan and one was a multisite study conducted in four Southern European countries. Chang (2015) was conducted in Antigua, Jamaica and St Lucia and is included because Antigua is classified as a high-income country.

Seventeen trials compared one intervention with usual care,[25–28 36 37 41 44 46 50 51 54–56 58 59 61] although minor augmentations to usual care were made in six of these (eg, with some other information or services made available to parents who wished to access them).[26 36 37 56 58 59] One of the trials compared two different interventions with usual care.[61] In the remaining five trials, two interventions were compared with each other.[24 30 47–49] The timing of intervention delivery varied, from the first month of life only[24] to longer term interventions, with eight studies including interventions that continued beyond the child's second birthday,[27 30 36 41 47 50 51 55] and the maximum intervention length being 5 years.[50] Studies ranged from 28 to 1593 participants: six included fewer than 100 participants; 12 included between 100 and 500; three included more than 500 participants; and one did not report the number of subjects recruited or analysed.[58] In 17 of the 22 trials, outcome data were available for 75% or more of those randomised. All of the trials offered coverage of the intervention to all families in the general population or within a neighbourhood or defined population (eg, recruitment occurred in hospitals serving areas with high levels of social disadvantage or the intervention was made available to all individuals within specific postcodes).[25 28 30 41 44 48 59 61] Three trials also included first time mothers only.

Six trials were classified as being at low risk of bias (all compared interventions with usual care), one was at high risk, and 15 had an unclear risk (figures 2 and 3).

## Intervention characteristics

Twenty eight interventions were examined in total (see table 2A for studies that included one intervention and table 2B for studies that compared two interventions). Most papers described the body of literature on which the intervention development had been based, but provided less detail on the proposed mechanisms of action of the intervention. Seven were of low intensity: short films followed by group discussions shown in health centre waiting rooms (Chang, see table 2A); sets of

building blocks and activity handouts sent to parents by post (Christakis, table 2A); 'literacy promoting anticipatory guidance' by paediatricians (High, table 2A); a brief parenting course (Hiscock, table 2A); access to community groups (Wiggins, intervention 2, table 2A) and two different methods for giving feedback to mothers on a neonatal behavioural assessment (Beeghly, table 2B). Ten were of moderate intensity. These included one-to-one home visits (between five (Cheng, table 2A) and twelve visits (Wiggins, table 2A) in total), group sessions (up to eight in total (Feinberg, Niccols 2008, Niccols 2009, all table 2A)), training for primary healthcare workers in interview techniques that encouraged consideration of child development (Tsiantis, table 2A), training for parents in daily activities to promote motor development (Lobo, table 2B) or a combination of different components (Santelices table 2A, Doyle table 2B). Eleven interventions were of high intensity. They were classified as such because they included multiple components (up to a maximum of eight) and regular contact with parents over a sustained period of time or intensive contact for a shorter period of time. In the five studies that included two interventions, the interventions were of the same intensity in all but one (Doyle, which compared a medium intensity intervention with one of high intensity). The aim of these studies was to compare different models of care with each other.

The mode of delivery of the intervention varied between trials. The intervention was delivered by health professionals in seven trials,[24–26 44 46 47 58] by other professionals (including 'parent educators', 'family visitors' or researchers) in eight trials,[30 36 48 50 54–56 59] by a mixture of health and other professionals in three trials[37 41 51] and by peer mentors in one trial.[28] One trial examined one intervention delivered by health professionals and another delivered by community support groups.[61] In the remaining trials, one included materials delivered to parents by post[27] and one examined training for parents by a physiotherapist to deliver a handling and positioning intervention.[49]

A full narrative summary of the results, including the tools used to assess the outcome in each trial and the estimates of intervention effects, is given in online supplementary web appendix C. Many of the trials reported several measures of the same outcome and/or measured outcomes at different time points, resulting in multiple comparisons for each outcome. The findings are summarised by outcome in table 3 and are described below. Effect estimates are given in the text below only for the studies found to be at low risk of bias. An effect direction plot[66] provides a visual display of the results across all outcome domains, ordered by risk of bias and the intensity of the intervention (table 4).

## Motor development outcomes

Six studies, including a total of 37 comparisons in 1276 participants, reported motor development outcomes using validated tools. The quality of the evidence was

**Table 1** Characteristics of the included studies

| Study setting | Type and aim of study | Comparison group | (1) Who received intervention; (2) When? | Sample size (1) randomised; (2) In analysis (% of randomised) | Universal or geographically targeted? | Outcome domains measured* |
|---|---|---|---|---|---|---|
| Beeghly[24] USA | Individual RCT to compare the effectiveness of two one-to-one clinic-based interventions (infant-centred vs mother centred) on motor and cognitive development post-intervention | Two interventions compared | (1) Mothers and infant; (2) When child was 3, 14 and 30 days of age | (1) 163; (2) 125 (77%) | Universal | Motor Cognitive |
| Chang et al[25] Antigua, Jamaica and St Lucia | Cluster RCT to examine the effectiveness of a group-based intervention on language and cognitive development post-intervention | Usual care | (1) Mothers; (2) When child was 3, 6, 9, 12 and 18 months | (1) 30 health centres randomised (501 women enrolled); (2) 426 (85%) | Geographically targeted | Motor Language Cognitive Overall |
| Cheng[26] Japan | Individual RCT to examine the effectiveness of an individual home-based intervention on social and emotional well being postintervention | Usual care, with a counselling service made available to all | (1) Mothers; (2) When child was 5-9 months | (1) 95; (2) 85 (89%) | Universal | SEWB |
| Christakis[27] USA | Individual RCT to examine the effectiveness of giving two sets of building blocks and a newsletter of activities to complete with them on language development and social and emotional well-being postintervention | Usual care | (1) Families; (2) When child was 18-30 months | (1) 175; (2) 140 (80%) | Universal | Language SEWB |
| Cupples[28] UK (Northern Ireland) | Individual RCT to examine the effectiveness of one-to-one contact with trained peer mentors on motor, cognitive development and social and emotional well being post-intervention | Usual care | (1) Mothers; (2) From 20 weeks of pregnancy to 12 months postpartum | (1) 343; (2) 280 (82%) | Geographically targeted (First-time mothers only) | Motor Cognitive SEWB |
| Doyle[30–35] Ireland | Individual RCT to compare the effectiveness of the 'high support' versus 'low support' versions of the multicomponent 'Preparing for Life' programme on motor, language and cognitive development and social and emotional well-being while intervention was ongoing | Two interventions compared | (1) Parents; (2) From pregnancy to school entry | (1) 233 (2) 173 (74% at 6 months) 165 (71% at 12 months) 154 (66% at 18 months) 166 (71% at 24 months) 151 (65% at 36 months) | Geographically targeted | Motor Language Cognitive SEWB Overall |
| Drotar[36] USA | Individual RCT to examine the effectiveness of the multi-component 'Born to Learn' programme on language and cognitive development and social and emotional well-being while intervention was ongoing | Usual care, plus handouts and offer of a different group meeting | (1) Parents and child; (2) Recruited between birth and 9 months; programme continued to age 3 | (1) 527; (2) 410 (78%, although inconsistent numbers presented in tables) | Universal | Language Cognitive† SEWB |
| Feinberg[37–40] USA | Individual RCT to examine the effectiveness of a group-based intervention ('Family Foundations') on social and emotional well-being postintervention | Usual care plus brochure on childcare options | (1) Parents; (2) Recruited during pregnancy, continued to age 4-6 months | (1) 169; (2) 152 (90% at 6 months); 154 (91% at 12 months); 137 (81% at 36 months) | Universal (First-time mothers only) | SEWB† |
| Griffith[41] UK (Wales) | Individual RCT to examine the effectiveness of a group-based parenting intervention ('Incredible Years Toddler Programme') on social and emotional well-being and overall development postintervention | Usual care (waiting list control group) | (1) Parents; (2) Children 12-36 months at baseline (mean age 21 months) | (1) 89; (2) 89 (100%) | Geographically targeted | SEWB Overall |
| High[44] USA | Individual RCT to examine the effectiveness of a one-to-one clinic-based intervention on language development postintervention | Usual care | (1) Parents; (2) Children 5-11 months at baseline | (1) 205; (2) 153 (75%) | Geographically targeted | Language |
| Hiscock[23 45 46] Australia | Cluster RCT to examine the effectiveness of a group-based intervention ('Toddlers Without Tears') on social and emotional well-being postintervention | Usual care | (1) Parents; (2) When child was 8, 12 and 15 months | (1) 40 maternal and child health centres randomised (733 women enrolled); (2) 672 (92% at 18 months); 656 (89% at 24 months); 589 (80% at 36 months) | Universal | SEWB |

Continued

**Table 1** Continued

| Study setting | Type and aim of study | Comparison group | (1) Who received intervention; (2) When? | Sample size (1) randomised; (2) In analysis (% of randomised) | Universal or geographically targeted? | Outcome domains measured* |
|---|---|---|---|---|---|---|
| Johnston[47] USA | Individual RCT to compare the effectiveness of the multicomponent 'Healthy Steps' programme with 'Healthy Steps' plus 'PrePare' on language development and social and emotional well-being postintervention | Two interventions compared | (1) Mothers; (2) Recruited during pregnancy, continued to age 3 | (1) 303; (2) 239 (79%) | Universal | Language SEWB |
| Landry[48] USA | Individual RCT to compare the effectiveness of two different models of home visits on language and cognitive development and social and emotional well-being postintervention | Two interventions compared | (1) Mothers; (2) When child was 6–10 months | (1) 264; (2) 240 (91%) | Geographically targeted | Language‡ Cognitive‡ SEWB‡ |
| Lobo[49] USA | Individual RCT to compare the effectiveness of a 'handling and positioning' intervention with a 'social interaction' intervention on motor development while the intervention was ongoing and postintervention | Two interventions compared | (1) Parents; (2) For 3 weeks, from when child was 2 months of age | (1) 28; (2) 28 (100%) | Universal | Motor |
| Miller[50] Ireland | Individual RCT to examine the effectiveness of the multi-component 'Lifestart' programme on cognitive development and social and emotional well being whilst intervention was ongoing | Usual care | (1) Parents; (2) Recruited when child < 12 months, continued to age 5 | (1) 435; (2) 347 (80% at 36 months) | Universal (although parents self-referred) | Cognitive SEWB |
| Minkovitz[51–53] USA | Individual RCT to examine the effectiveness of the multi-component 'Healthy Steps' programme on social and emotional well-being in the long-term while intervention was ongoing | Usual care | (1) Families; (2) Recruited at birth or first well-child visit, continued to age 3 | (1) 2235; (2) 1593 (71%) | Universal | SEWB |
| Niccols[54] Canada | Individual RCT to examine the effectiveness of a group-based intervention ('Right from the Start') on social and emotional well-being postintervention | Usual care | (1) Mothers; (2) Children 1–24 months at baseline | (1) 76; (2) 73 (96% immediately postintervention) 64 (84% at 6 months) | Universal (although parents self-referred) | SEWB |
| Niccols[55] Canada | Individual RCT to examine the effectiveness of a group-based intervention ('COPEing with Toddler Behaviour') on social and emotional well-being postintervention | Usual care (waiting list control group) | (1) Mothers; (2) Children 12–36 months at baseline (mean age 24 months) | (1) 79; (2) 74 (94% immediately postintervention) 71 (90% at 1 month) | Universal (although parents self-referred) | SEWB† |
| Santelices[56] Chile | Individual RCT to examine the efficacy of a multicomponent intervention ('Promoting Secure Attachment') on social and emotional well-being postintervention | Usual care, plus one lecture by a psychologist | (1) Mothers; (2) Recruited during late pregnancy, continued to age 1 | (1) 100; (2) 72 (72%) | Universal (First-time mothers only) | SEWB |
| Tsiantis[57 58] Cyprus, Greece, Portugal, Yugoslavia | Cluster RCT to examine the effectiveness of training primary healthcare workers to use semistructured interviews to promote language development and social and emotional well-being while intervention is ongoing | Usual care (healthcare workers in this group received one lecture) | (1) Mothers; (2) Recruited during pregnancy, continued to age 2 | (1) 80 primary healthcare workers (number of women randomised not reported, 'recruitment did not achieve target figures'); (2) Not reported | Universal | Language SEWB† |
| Wagner[59 60] USA | Individual RCT to examine the effectiveness of a multicomponent intervention ('Parents as Teachers') on motor, language and cognitive development and social and emotional well-being while intervention is ongoing | Usual care, plus age-appropriate toys 'at regular intervals' and an annual child assessment | (1) Mothers; (2) Recruited during pregnancy, continued to age 2 | (1) 665; (2) 266 (40%) | Geographically targeted | Motor Language Cognitive SEWB |

Continued

**Table 1** Continued

| Study setting | Type and aim of study | Comparison group | (1) Who received intervention; (2) When? | Sample size (1) randomised; (2) In analysis (% of randomised) | Universal or geographically targeted? | Outcome domains measured* |
|---|---|---|---|---|---|---|
| Wiggins[61 62] UK (England) | Individual RCT to examine the effectiveness of two postnatal social support interventions (SHV and CGS) on language development, social and emotional well-being overall development postintervention | Both interventions compared with usual care | (1) Mothers; (2) Recruited when child was ~10 weeks, continued to age 1 | (1) 731 (SHV 183, CGS 184, control 364); ii) SHV comparison: 493 (91% at 12 months); 443 (81% at 18 months). CGS comparison: 492 (90% at 12 months); 456 (83% at 18 months). | Geographically targeted | Language§ SEWB§ Overall§ |

*Used a validated questionnaire for measuring outcome unless indicated otherwise (although the use of the instrument may not always have been validated in the target population).
†Used a combination of validated questionnaires and coding of videotaped activities and behaviours (no validated coding framework described).
‡Used coding of videotaped activities and behaviours (no validated coding framework described).
§No validated measure used; asked parents whether they perceived their child's development to be normal and whether they had worries about specific areas of development (including speech and behaviour).
CGS, Community Group Support; RCT, randomised controlled trial; SEWB,social and emotional well-being; SHV, Support Health Visitors.

moderate. Three studies comparing one intervention with usual care showed no effect (972 participants, three comparisons), and three studies in which two interventions were compared (304 participants) showed no effect in 25 of 34 comparisons. The positive effects were all from one study of 28 infants who received a daily 15-minute handling and positioning intervention or a social interaction intervention for three weeks.[49] In addition to its small sample size, this study had an unclear risk of bias. Only one study at low risk of bias examined motor development outcomes.[28] This study found no difference in the mean scores for the psychomotor scores of the Bayley Scale of Infant Development between the intervention and control groups (mean difference 1.64, 95% CI -0.94 to 4.21, p=0.21).

### Language development outcomes

Ten studies including a total of 43 comparisons examined this outcome, with nine using validated tools. The total number of participants for this outcome is unknown as one study did not report numbers,[58] but was in excess of 3000. The quality of the evidence was low. Seven studies comparing 1 intervention with usual care showed no effect in 10 comparisons, a positive effect in 4 comparisons and a negative effect (poorer language development in the intervention group) in 2 comparisons. Three studies which compared 2 interventions (632 participants) found no difference between the interventions in 23 comparisons and a positive effect in 4 comparisons. Only one study at low risk of bias examined language development outcomes.[61] This study compared two different interventions with usual care (supportive health visiting (SHV; moderate intensity) and community groups (CGS; low intensity)). This study found that fewer mothers in the SHV group expressed a worry about their child's speech than in the control group (risk ratio 0.46, 95% CI 0.23 to 0.93), and no difference in the number of number of mothers expressing worries about speech between CGS and control (risk ratio 1.22, 95% CI 0.78 to 1.92).

### Cognitive development outcomes

Eight studies, including a total of 40 comparisons in 2245 participants, examined cognitive development outcomes. All used validated tools, except for one where videotaped interactions were coded for 'independent goal-directed play'.[48] The quality of the evidence was low. In 5 studies (1729 participants) comparing 1 intervention with usual care, there was no effect in 18 of 20 comparisons, and a positive effect in 2 comparisons. Three studies which compared two interventions (516 participants) found no difference between the interventions in 16 of 20 comparisons and a positive effect in four comparisons. Two studies at low risk of bias examined cognitive development outcomes. One study[28] found no difference in the mean scores in the intervention and control group for the mental development scores of the Bayley Scale of Infant Development (mean difference −0.81, 95% CI −2.81 to 1.16, p=0.42). The other[50] found no difference

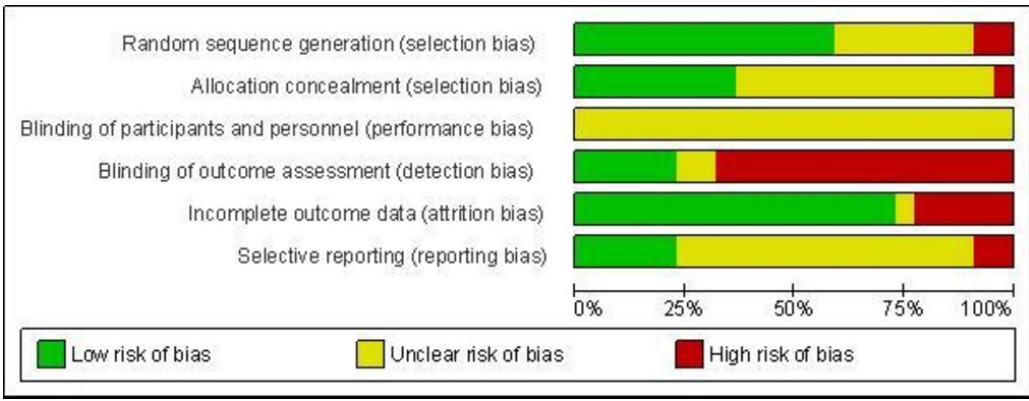

**Figure 2** Risk of bias graph: review authors' judgements about each risk of bias item presented as percentages across all included studies.

in mean scores between intervention and control on cognitive development using the British Ability Scale (mean score in intervention group -0.05 (SD 1.01) and in control group 0.03 (SD 0.99), Hedges g effect size −0.63, 95% CI −0.28 to 0.15, p=0.56).

## Social and emotional well-being outcomes

These outcomes were examined in 156 comparisons in 18 trials (total participant numbers unknown as 1 study did not report participant numbers[58] but was in excess of 5000). Many different outcomes were examined (see online table C4 in the supplementary web appendices for details), with most assessed using validated tools (such as the Child Behaviour Checklist, the Infant Behaviour Questionnaire, the Parent–Infant Relationship Global Assessment, the Q-Sort Measure of the Security of Attachment and social and emotional well-being scores from the Ages and Stages Questionnaire). Most focused on behavioural outcomes.

The quality of the evidence was low. In 15 studies comparing 1 intervention with usual care, there was no effect in 60 of 69 comparisons. In the 3 studies which compared 2 interventions (630 participants), there was no difference between the interventions in 82 of 87 comparisons, a positive effect in four comparisons and a negative effect in one comparison. Six studies at low risk of bias examined social and emotional well-being outcomes, and none found a difference between intervention and control groups. The largest of these[46] found no difference in mean scores between intervention and control for externalising or internalising behaviours measured using the Child Behaviour Checklist at 3, 9 or 21 months postintervention. For example, the adjusted mean difference for externalising behaviours at 3 months was 0.16 (95% CI −1.01 to 1.33, p=0.79), at 9 months was −0.79 (95% CI −2.27 to 0.69, p=0.30) and at 21 months was −0.80 (95% CI −2.2 to 0.6, p=0.26).

## Overall child development outcomes

Four studies including a total of 12 comparisons in 1565 participants examined global estimates of child development. The quality of the evidence was moderate. Three

studies (1414 participants) comparing one intervention with usual care found no effect in seven of eight comparisons based on validated measures of global child development (Griffith Mental Development Scale[25] and mean score from the Schedule of Growing Skills II[41]). Two studies at low risk of bias examined this outcome. In one study,[61] there was no difference between SHV and control (risk ratio 0.88, 95% CI 0.39 to 1.99) or CGS and control (risk ratio 0.57, 95% CI 0.22 to 1.52) in the mother's perception of whether her child's development was normal. However, mothers in the SHV group had fewer mean number of worries about their child's development than in the control group (mean difference −0.23, 95% CI −0.42 to −0.01), but there was no difference in the mean number of worries about their child's development between CGS and control (mean difference 0.13, 95% CI −0.10 to 0.36). The other study, comparing 2 interventions (151 participants), found no difference between the interventions in 4 comparisons (using the mean development score from the Ages and Stages Questionnaire).[30]

## Subgroup effects reported within studies

Subgroup comparisons presented within the individual studies included examining whether the effects were different in families of different incomes or in children with different characteristics (eg, low birthweight infants vs normal birthweight infants, see tables 3 and 4). Some positive effects were seen, but the reporting of these analyses was generally incomplete, with an emphasis on positive intervention effects. No conclusions can therefore be drawn on subgroups in this review.

## Stratification of results across studies by risk of bias and intensity of interventions

Table 4 gives the effect direction plot, summarising the results for each outcome, ordered by risk of bias and the intensity of the intervention. In the studies at low risk of bias, there was no intervention effect when either low or high-intensity interventions were studied. Some positive effects were seen in the two trials of moderate intensity interventions, although in one, this was limited to subgroups only (children with 'disturbed' attachment

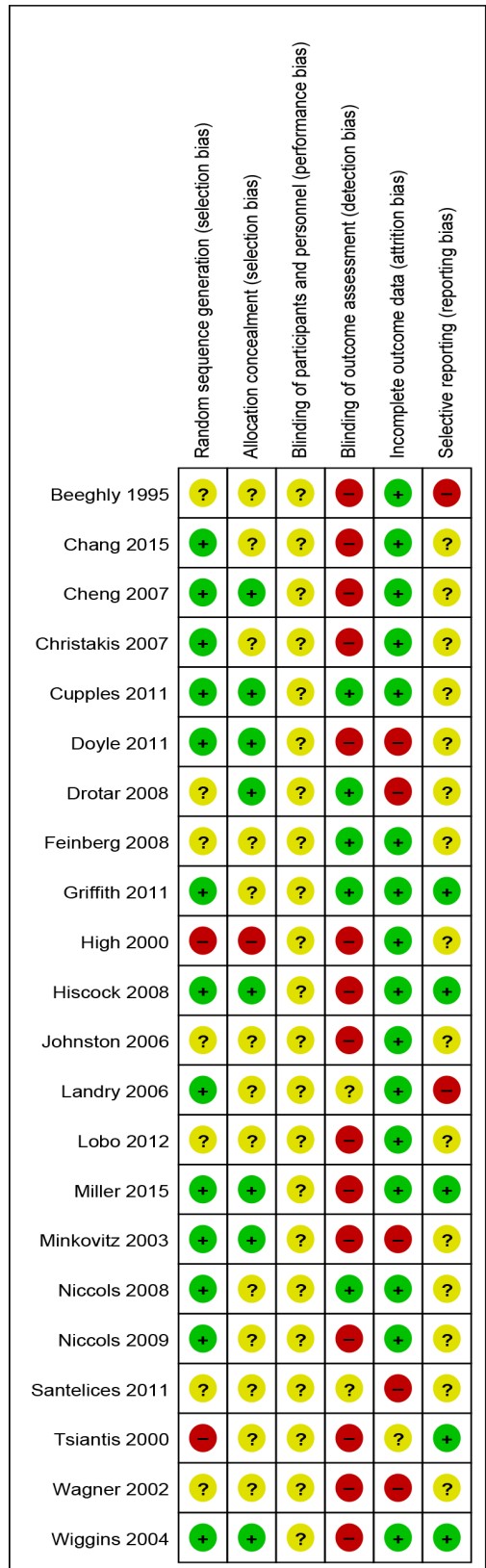

**Figure 3** Risk of bias summary: review authors' judgements about each risk of bias item for each included study.

at baseline),[26] and in the other, positive effects were not consistently seen.[61]

One study was classified as being at high risk of bias, and this examined a low intensity intervention.[44]

Inconsistent positive intervention effects were seen in this study, with most of these in one subgroup only. The remaining studies were classified as being at unclear risk of bias, and there is no clear pattern the effects seen in these studies. Programme intensity does not appear to be associated with effectiveness in these studies, in that there is no evidence that higher intensity interventions are associated with more intervention effects.

Table 4 also summarises the uptake and adherence to intervention components. These factors were variable across studies and inconsistently reported. For example, for low-intensity interventions, this ranged from only 19% of the women accessing the intervention at all (community support groups[61]) to 83% accessing every session.[25] Patterns of adherence to the moderate and high-intensity interventions also varied.

No clear pattern in the results were seen when stratification by the other prespecified variables was conducted (see online supplementary web appendix D).

## DISCUSSION

The need for interventions to promote child development outcomes in all families has been clearly articulated. Using a broad systematic search of the extensive literature in this field, we found 22 RCTs examining the effect of interventions that enhance health service contacts from the antenatal period to 24 months postpartum. The interventions varied greatly in their content and intensity, and uptake, adherence and fidelity were not consistently reported. The quality of evidence for motor development and overall child development was moderate, and the majority of comparisons showed no intervention effect. The quality of evidence for language development, cognitive development and social and emotional well-being was low. The majority of the comparisons for these outcomes showed no effect, and where positive impacts were observed, within-study effects were inconsistent. Studies that compared one intervention with usual care did not demonstrate more positive intervention effects than studies comparing two interventions. We conclude that there is insufficient evidence to suggest that the interventions reviewed here are effective at improving child development outcomes. The low-to-moderate quality of evidence overall suggests that there is a need for high-quality robust trials to inform current health service delivery in this area.

The strength of our review was the broad search strategy, which encompassed many sources of information other than database searching. We are confident that we have identified most relevant studies (including three trials not yet published in peer-reviewed journals). Although it was not possible to conduct a meta-analysis due to the variation in the types of interventions and methods used to measure outcomes, the narrative review—supplemented with the effect direction plot—provides a comprehensive picture of the limited evidence-base in this field.

To our knowledge, this is the first systematic review of interventions which aim to enhance health service contacts

**Table 2** Description of intervention components and intensity

| Study | Description | 1 | 2 | 3 | 4 | 5 | 6 | 7 | 8 | 9 | 10 | Contacts: Number Frequency Duration | Who delivered the intervention? |
|---|---|---|---|---|---|---|---|---|---|---|---|---|---|
| A. Studies comparing one intervention with usual care | | | | | | | | | | | | | |
| *Low intensity* | | | | | | | | | | | | | |
| Chang[25] | Three 3-min films demonstrating 'behaviours central to promoting child development' shown as women waited for 3, 6, 9, 12 and 18 month vaccine visits, followed by group discussions with a community health worker. Cards given to reinforce messages, plus picture book at 9 and 12 months and puzzle at 18 months. | | | | ✓ | | | ✓ | | | ✓ | 5 Every 3–6 months Over 15 months | Health professionals |
| Christakis[27] | Parents were sent two sets of building blocks with accompanying newsletters containing suggested activities in the post. | | | | ✓ | ✓ | | ✓ | | | | 2 Bimonthly 3 months | No contact with families postrecruitment |
| High[44] | Paediatricians gave books, handouts and 'literacy promoting anticipatory guidance' to parents at routine well-child visits. | | ✓ | | ✓ | | | ✓ | | | | 5 Every 3 months Over 12 months | Health professionals |
| Hiscock[23 45 46] | 'Universal anticipatory guidance' with strategies for behavioural difficulties: handout at 8 months; two 2-hour group sessions at 12 and 15 months. | | | ✓ | ✓ | | | | | | | 3 Every 3–4 months Over 7 months | Health professionals |
| Wiggins[61 62] Intervention 2 | Access given to mothers to community group support that already existed and which provided drop in sessions and/or telephone support and/or home visits (different services provided by each of the 8 groups who agreed to take part in the study); participants selected whether to make contact and attend groups. | ✓ | | ✓ | | | | | ✓ | | | Variable | Other professionals |
| *Moderate intensity* | | | | | | | | | | | | | |
| Cheng[26] | Five one-to-one home visits of one hour 'aimed at improving the quality of mother–infant relationship'; tailored encouragement and advice given following observation of mothers playing with infants. | ✓ | | | | | | | | | | 5 Monthly Over 5 months | Health professionals |
| Feinberg[37–40] | Four prenatal and four postnatal interactive group sessions, designed 'to enhance coparenting'. | | | ✓ | | | | | | | | 8 Every 6 weeks Over 11 months | Mix of health and other professionals |
| Niccols[54] | Eight 2-hour group sessions using a 'coping modeling problem solving approach', to enhance caregiver skills in 'reading infant cues and responding sensitively' plus homework. | | | ✓ | | | | | | | | 8 Weekly Over 2 months | Other professionals |
| Niccols[55] | Eight 2-hour group sessions, using a 'coping modeling problem solving approach', to train parents on effective parenting styles and strategies, plus homework. | | | ✓ | | ✓ | | | | | | 8 Weekly Over 2 months | Other professionals |
| Santelices[56] | Six 2-hour group sessions during pregnancy on 'maternal sensitivity…and to promote the development of a secure and healthy bond between mother and child', and 4-hour long one-to-one sessions postpartum to observe interactions and give feedback. | ✓ | ✓ | ✓ | | | | | | | | 10 Variable Over 16 months | Other professionals |
| Tsiantis[57 58] | Primary healthcare workers trained to use a semistructured interview technique during six to eight routine visits to discuss age-appropriate child development topics. | | ✓ | | | | | | | | | 6–8 Variable Over 36 months | Health professionals |
| Wiggins[61 62] Intervention 1 | Supportive home visits conducted postnatally by five very experienced health visitors, adapted to each woman's needs to address her concerns and questions | ✓ | | | | | | | | | | 12 Monthly Over 12 months | Health professionals |
| *High intensity* | | | | | | | | | | | | | |

**Table 2** Continued

| Study | Description | 1 | 2 | 3 | 4 | 5 | 6 | 7 | 8 | 9 | 10 | Contacts: Number Frequency Duration | Who delivered the intervention? |
|---|---|---|---|---|---|---|---|---|---|---|---|---|---|
| | | | | | | Components* | | | | | | | |
| Cupples[28] | Trained peer mentors provided one-to-one support on 'health-related' topics via home visit or phone call | ✓ | | | | | | | | | | 22 Every 2-4 weeks Over 17 months | Peer mentors |
| Drota[36] | One-to-one home visits; monthly parent group sessions; annual developmental and health screening; access to resource network | ✓ | | ✓ | | | ✓ | | ✓ | ✓ | | 27 Every 2-4 weeks Over 36 months | Other professionals |
| Griffith[41] | 12 2-hour group sessions including watching videos, group discussions and role play to help understand and manage child behaviour; homework tasks to complete. | | | ✓ | | ✓ | | | | | | 12 Weekly Over 3 months | Mix of health and other professionals |
| Miller[50] | 30-60 min monthly home visit by a Lifestart family visitor and a monthly magazine ('Growing Child') of age-appropriate activities | ✓ | | | ✓ | | | | | | | 35 Monthly Over 36 months | Other professionals |
| Minkovitz[51-53] | Enhanced well-child care (12 visits, including access to Reach Out and Read literacy programme), 6 home visits in 3 years; telephone line; developmental screening; written guidance; monthly parent groups; links to community resources | ✓ | ✓ | ✓ | ✓ | | ✓ | | ✓ | ✓ | | 16 Variable Over 36 months | Mix of health and other professionals |
| Wagner[59 60] | Monthly home visits and parent group meetings to provide information on child development and demonstrate age-appropriate activities. Periodic developmental screening and, if needed, referrals to community services provided. | ✓ | | ✓ | | | ✓ | | | ✓ | | 24 Monthly Over 24 months | Other professionals |
| **B. Studies comparing two interventions with each other** | | | | | | | | | | | | | |
| Beeghly[24] 1: Low intensity | Three individual 45-min sessions where mother observed a NBAS and discussed findings with a paediatrician (including exploring the caregiving that might promote the mother-child relationship) | | ✓ | | | | | | | | | 3 At 3, 14 and 30 days old | Health professionals |
| Beeghly[24] 2: Low intensity | Three individual 45-min sessions, where mother discussed her perceptions of motherhood and concerns with a paediatrician and was given feedback about an NBAS that was conducted in a different room. | | ✓ | | | | | | | | | 3 At 3, 14 and 30 days old | |
| Lobo[49] 1: Moderate intensity | Parents taught a positioning and handling programme during a home visit by a physiotherapist to be completed for 15 min daily for 3 weeks. Six assessment home visits also completed. Caregivers given manual and a session diary. | ✓ | | | ✓ | ✓ | | | | | | 6 Every 2 weeks Over 3 months | Intervention delivered by parents after training |
| Lobo[49] 2: Moderate intensity | Parents asked to engage their child in 15 min of face-to-face interaction daily for 3 weeks. This group also had 6 assessment visits. | ✓ | | | | | | | | | | 6 Every 2 weeks Over 3 months | |
| Doyle[30-35] 1: Moderate intensity | Access to a support worker; annual packs containing toys and books (worth €100); facilitated access to 1 year of preschool; stress control and healthy eating sessions | | ✓ | ✓ (2) | ✓ | | | ✓ | | ✓ | ✓ | Variable Variable Over 36+ months | Other professionals |
| Doyle[30-35] 2: High intensity | Home visits from a trained mentor; tip sheets; Triple P Positive Parenting group sessions; baby massage; annual packs containing toys and books (worth €100); facilitated access to one year of preschool; stress control and healthy eating sessions | ✓ | | ✓ (3) | ✓ | | | ✓ | | ✓ | ✓ | Variable Weekly Over 36+ months | |
| Johnston[47] Intervention 1: High intensity | Enhanced well-child care (six visits, including Reach Out and Read literacy programme); six home visits in 3 years; telephone line; developmental screening; written guidance; monthly parent groups; links to community resources | ✓ | ✓ | ✓ | ✓ | | ✓ | | | ✓ | | Variable Monthly Over 36+ months | Health professionals |
| Johnston[47] Intervention 2: High intensity | As above, plus three additional home visits during second half of pregnancy | ✓ | ✓ | ✓ | ✓ | | ✓ | | ✓ | ✓ | | Variable Monthly Over 36+ months | |

Continued

**Table 2** Continued

| Study | Description | Components* | | | | | | | | | | Contacts: Number Frequency Duration | Who delivered the intervention? |
|---|---|---|---|---|---|---|---|---|---|---|---|---|---|
| | | 1 | 2 | 3 | 4 | 5 | 6 | 7 | 8 | 9 | 10 | | |
| Landry[48] Intervention 1:High intensity | Playing and Learning Strategies: one-to-one home visits of 1.5 hours to discuss the child's current development and behaviour, feedback on videotaped interactions with child; and planning with mothers of how to increase their 'responsive' behaviours | ✓ | | | | | | | | | | 10 Weekly Over 3 months | Other professionals |
| Landry[48] Intervention 2:High intensity | Developmental assessment screening: one-to-one home visits of 1.5 hours consisting of developmental screening and discussions on child development. Handouts on common issues (eg, sleep, feeding) given. | ✓ | | | ✓ | | ✓ | | | | | 10 Weekly Over 3 months | |

*1=one-to-one home visits; 2=one-to-one clinic visits; 3=group sessions; 4=handouts; 5=activities to perform at home; 6=developmental screening; 7=toys and/or books;8=telephone support; 9=access to community resources; 10=other.
NBAS, Neonatal Behavioral Assessment.

to improve child development outcomes, including social and emotional well-being outcomes in the very early years. Our conclusion is consistent with other reviews of early years interventions. For example, the Allen review[9] found that none of the interventions designed for universal use in the early years (defined as conception to school) had 'best' quality evidence available to support them. A recent rapid review to update the evidence for components of the Healthy Child Programme in England also found few studies of interventions aiming to promote child development outcomes in all families with children in the 0–5 age range.[10] We reviewed a larger number of primary studies than either of these previous publications. Previous studies have also examined the effects of programmes such as these on parental knowledge, attitudes or practices. We did not systematically review parental outcomes here, so cannot comment on whether parents benefited from these interventions. However, we can conclude that—in these studies—any effects on the parents did not, in turn, lead to consistent improvements in child development outcomes.

Understanding how health service contacts can be enhanced to provide support for parents to achieve the best possible developmental outcomes for their children is necessary but challenging. Maternal and child health services consist of many components, many of these untested. Parents also access a wide variety of other forms of support, and the effects of these are poorly understood. Although the evidence base examined in this review is limited, it does allow us to conclude that there is no convincing evidence that the interventions studied provide an additional benefit to the care currently provided in the settings included in these trials. There was also no evidence that interventions of high intensity confer more benefit than those of lower intensity as no dose–response relationship was evident: programmes of greater intensity (in terms of length, number or type of components) did not show more positive intervention effects than programmes of lower intensity. This is consistent with recent evidence for targeted interventions (such as the recent trial of the Family Nurse Partnership programme in the UK[67]) and has implications for commissioners of early years health services.

Many interventions currently incorporated into health services have not been adequately evaluated, and we recommend further research to generate this evidence. The methodological quality of many of the studies—or the reporting of their methods—was poor (as shown in figure 2 and 3). Eight of 22 trials provided no detail on how their randomisation sequence was generated, and one reported using an inappropriate method. Thirteen provided no detail of allocation concealment, and one reported using an inappropriate method. Ten relied on parental reporting of outcomes only, and a further five used a mix of parental reporting and observations. Although blinding of outcome assessment can be a challenge in studies that rely on parental reporting of their child's development, validated measures of assessing children's development without using parental report (eg, coding of videotaped interactions as used in [26 54 55 58]) exist and we would encourage their use

**Table 3** Summary of findings

Population: Parents from antenatal period to 2 years postpartum
Settings: Universal programmes offered within defined populations
Intervention: Programmes to improve child development outcomes by enhancing health professional contact
Comparison: Usual care, or two different interventions compared

| Number of studies Total number of participants* Total number of comparisons† | Results | Quality of the evidence |
|---|---|---|
| **Outcome: Motor development** | | |
| Comparison: Intervention with usual care | | |
| 3 studies[22 25 56] 972 participants, 3 comparisons | No effect in the three comparisons. No effect in the two subgroups examined in one study. | Moderate (downgraded one level because of risk of bias) |
| Comparison: Two interventions | | |
| Three studies[21 27 46] 304 participants, 34 comparisons (20 in Doyle, 13 in Lobo) | No difference in 25 comparisons; better outcomes in the more intensive intervention group in nine comparisons (all in one study). No difference in the two subgroups examined in one study. | Moderate (downgraded one level because of risk of bias) |
| **Outcome: Language development** | | |
| Comparison: Intervention with usual care | | |
| Seven studies†[22 24 33 41 54 56 58] 16 comparisons (6 in High) Participant numbers not known§, | No effect in 10 comparisons; better outcomes in intervention than control in four comparisons (two studies); worse outcomes in intervention than control group in two tests (one study). Subgroup effects reported in 4 studies, with some better outcomes in intervention than control, but reporting of subgroups unclear and incomplete. | Low (downgraded two levels because of risk of bias and inconsistency)¶ |
| Comparison: Two interventions | | |
| Three studies[27 44 45] 632 participants, 27 comparisons (21 in Doyle) | No difference between the two interventions in 23 comparisons; better outcomes in the more intensive intervention group in four tests (two studies). No subgroup effects examined. | Low (downgraded two levels because of risk of bias and inconsistency)¶ |
| **Outcome: Cognitive development** | | |
| Comparison: Intervention with usual care | | |
| Five studies[22 25 33 47 56] 1729 participants, 20 comparisons (16 in Drotar) | No effect in 18 comparisons; better outcomes in intervention than control in two comparisons (two studies). Subgroup effects examined in two studies, with some better outcomes in intervention than controls seen, but reporting of subgroups unclear and incomplete. | Low (downgraded two levels because of risk of bias and inconsistency)¶ |
| Comparison: Two interventions | | |
| Three studies[21 27 45] 516 participants, 20 comparisons (18 in Doyle) | No difference between the two interventions in 16 comparisons; better outcomes in the more intensive intervention group in four comparisons (two studies). No significant interaction in the four tests performed in one study. | Low (downgraded two levels because of risk of bias and inconsistency)¶ |
| **Outcome: Social and emotional well being** | | |
| Comparison: Intervention with usual care | | |
| 15 studies†[23–25 33 34 38 43 47 49 51–54 56 58] 69 comparisons (7 in Drotar, 14 in Feinberg, 8 in Niccols, 16 in Tsiantis, 5 in Wagner) Participant numbers not known§, | No effect in 60 comparisons (in one study**); better outcomes in intervention than control in nine comparisons (two studies). Subgroup effects examined in four studies, with some better outcomes in intervention than control seen, but reporting of subgroups unclear and incomplete. | Low (downgraded two levels because of risk of bias and inconsistency) |
| Comparison: Two interventions | | |
| Three studies[27 44 45] 630 participants, 87 comparisons (78 in Doyle, 7 in Landry) | No difference between the two interventions in 82 comparisons; better outcomes in one intervention compared with another in four comparisons (two studies); worse outcome in the more intensive intervention group in one test (in one study). Subgroup effects examined in one study, with some better outcomes in one intervention compared with another, but reporting of subgroups unclear and incomplete. | Low (downgraded two levels because of risk of bias and inconsistency) |

Continued

**Table 3** Continued

**Population: Parents from antenatal period to 2 years postpartum**
**Settings: Universal programmes offered within defined populations**
**Intervention: Programmes to improve child development outcomes by enhancing health professional contact**
**Comparison: Usual care, or two different interventions compared**

| Number of studies Total number of participants* Total number of comparisons† | Results | Quality of the evidence |
|---|---|---|
| Outcome: Overall child development | | |
| Comparison: Intervention with usual care | | |
| Three studies‡[22 38 58] 1414 participants, 8 comparisons | No effect in seven comparisons; better outcomes in intervention than control in one comparison (one study). No subgroups examined. | Moderate (downgraded one level because of inconsistency) |
| Comparison: Two interventions | | |
| One study[27] 151 participants, 4 comparisons | No difference between the two interventions in the four comparisons. | Moderate (downgraded one level because of risk of bias) |

GRADE Working Group grades of evidence.
High quality: Further research is very unlikely to change our confidence in the summary of the effects.
Moderate quality: Further research is likely to have an important impact on our confidence in the summary of the effects.
Low quality: Further research is very likely to have an important impact on our confidence in the summary of the effects.
Very low quality: We are very uncertain about the summary of the effects.
*As the number of participants can vary within studies (eg, where outcomes are measured at several different time points), the total number of participants noted here is calculated from the total numbers analysed at the last time point in each study; this is therefore a conservative estimate of the total number of participants for each outcome.
†Total number of comparisons performed for the specified outcome in the whole study population across all of the studies; studies in which five or more comparisons are conducted on the same outcome (either at one time point or across different time points) are referenced.
‡Includes both comparisons in Wiggins (supportive health visiting with usual care and community groups with usual care)
§Total participant numbers not reported in Tsiantis.
¶Inconsistency noted where (1) positive, negative and no effects are reported for an outcome and/or (2) there is a different effect seen in more than 30% of comparisons across studies and/or (3) different effects are reported within a study and/or (4) most of the positive or negative effects are seen in subgroups only and the reporting of subgroup effects is incomplete or inconsistent
**The comparison reported is the ratio or difference in estimates between intervention and control group or between the two intervention groups, at the specified follow-up point unless otherwise noted (**indicates where the difference in change between intervention and control is used instead).

**Table 4** Effect direction plot, ordered by risk of bias and intensity of intervention (key given in footnote)

A. Studies comparing one intervention with usual care

| Study Intervention intensity | Study design | Risk of bias | Adherence | Timing | When? | Motor | Lang | Cogn | SEWB | Overall | Additional detail on intervention effects or subgroup analyses |
|---|---|---|---|---|---|---|---|---|---|---|---|
| Hiscock[23 45 46] Low | cRCT | Low | 49% of parents attended all sessions | Post | Short | | | | ○ | | |
| | | | | Post | Medium | | | | ○ | | |
| | | | | Post | Long | | | | ○ | | |
| Wiggins[61 62] Low (CGS) | iRCT | Low | 19% of women attended a group | Post | Immed | | | | | ○ | For the short-term analysis, only subgroup analyses (by attachment quality) presented; results inconsistent. |
| | | | | Post | Short | | ○ | | ○ | ○ | |
| Cheng[26] Moderate | iRCT | Low | Not reported | Post | Short | | | | ○ | | |
| | | | | Post | Long | | | | | | |
| Wiggins[61 62] Moderate (SHV) | iRCT | Low | Mean number of visits=7 (of 12 planned) | Post | Immed | | ● | | ○ | ○ | One of one comparison showed improved language outcome and one of one comparison showed improved overall development in intervention group. |
| | | | | Post | Short | | ● | | ○ | ● | |
| Cupples[28] High | iRCT | Low | Mean number of contacts=8.5 (of 22 planned) | Post | Immed | ○ | | | | ○ | |
| Griffith[41] High | iRCT | Low | 60% attended 8 or more sessions (of 12 planned) | Post | Short | | ○ | | ○ | ○ | |
| Miller[50] High | iRCT | Low | Adherence data currently being analysed | During | Long | | | ○ | ○ | | |
| High[44] Low | iRCT | High | Mean number of visits=3.4 (of 5 planned) | Post | Short | | ○ | | ○ | ○ | Three of six comparisons showed improved language outcomes in intervention group. Subgroup results: no differences seen in 13–17-month olds; 6 of 6 comparisons in 18–25-month olds showed improved language outcomes in intervention group; no test for interaction presented. |
| Chang[25] Low | cRCT | Unclear | 83% of mothers attended all visits | Post | Short | ○ | ○ | ○ | ○ | ○ | Improved cognitive outcome in intervention group on adjusting for potential confounders. |
| Christakis[27] Low | iRCT | Unclear | Not reported | Post | Short | | ○ | | ○ | ○ | Subgroup results: two of three comparisons in low income group showed improved SEWB outcomes in intervention group; test for interaction not presented |
| Feinberg[37–40] Moderate | iRCT | Unclear | 80% attended at least 3 of 4 antenatal sessions; 60% attended at least 3 of 4 postnatal sessions | Post | Short | | | | ○ | | Three of five comparisons in short term and one of two comparisons in medium term showed improved SEWB outcomes in intervention group. Results presented in text suggest there may be interaction effects with gender, but there is incomplete reporting of the subgroup analyses. |
| | | | | Post | Medium | | | | ○ | | |
| | | | | Post | Long | | | | ○ | | |
| Niccols[54] Moderate | iRCT | Unclear | 58% attended 4 or more sessions (of 8 planned) | Post | Immed | | | | ○ | | |
| | | | | Post | Short | | | | ○ | | |
| Niccols[55] Moderate | iRCT | Unclear | Not reported | Post | Immed | | | | ○ | | Two of four comparisons immediately postintervention term showed improved SEWB outcomes in intervention group. Three of four comparisons showed improved SEWB outcomes in intervention group in short term. |
| | | | | Post | Short | | | | ● | | |
| Santelices[56] Moderate | iRCT | Unclear | Not reported | Post | Short | | | | ○ | | |
| Tsiantis[57 58] Moderate | cRCT | Unclear | Not reported | Post | Short | | | | ○ | | Two of two comparisons showed poorer language outcomes in the intervention arm in the long term. One of two comparisons showed poorer SEWB outcomes in the intervention arm in the medium term. One of eight comparisons showed improved SEWB outcomes in the intervention arm in the long term. Incomplete reporting of the subgroup analyses. |
| | | | | Post | Medium | | | | ○ | | |
| | | | | Post | Long | | Ø | | | | |
| Drotar[36] High | iRCT | Unclear | Not reported | During | Short | | | | ○ | | 1 of 12 comparisons showed improved SEWB outcomes in the intervention arm in the long term. Incomplete reporting of the subgroup analyses. |
| | | | | During | Medium | | | ○ | ○ | | |
| | | | | During | Long | | ○ | ○ | ○ | | |
| Minkovitz[51–53] High | iRCT | Unclear | 79% of parents received 4 or more services (of 16) | During | Long | | | | ○ | | |

Continued

**Table 4** Continued

| Study Intervention intensity | Study design | Risk of bias | Adherence | Outcome measurement Timing | When? | Motor | Lang | Cogn | SEWB | Overall | Additional detail on intervention effects or subgroup analyses |
|---|---|---|---|---|---|---|---|---|---|---|---|
| Wagner[59 60] High | iRCT | Unclear | 44% of families still receiving services at 2 years | During | Long | ○ | ○ | ○ | ○ | ○ | Results also stratified by income; no significant interactions reported. |
| **B. Studies comparing two interventions** | | | | | | | | | | | |
| Beeghly[24] 2 low intensity | iRCT | Unclear | Not reported | Post | Short | ○ | | ○ | | | Tested for interaction between intervention and parity, IUGR, 'demographic' risk and maternal psychological risk; no significant interactions found. |
| Lobo[49] 2 moderate intensity | iRCT | Unclear | Excluded individuals who did not perform intervention on at least 60% of expected days | During | Short | ○ | | | | | Four of eight comparisons showed improved motor outcomes in the intervention arm in the short term, while intervention was ongoing. Five of five comparisons showed improved motor outcomes in the intervention arm postintervention. |
| | | | | Post | Short | ● | | | | | |
| Doyle[30–35] 1 high and 1 moderate | iRCT | Unclear | High: Mean number of visits = 46 Moderate: Not reported | During | Short | | ○ | ○ | ○ | ○ | 3 of 12 comparisons showed improved cognitive outcomes, and 1 of 62 comparisons showed improved SEWB outcomes in the intervention arm, in the long term, while intervention was ongoing. |
| | | | | During | Medium | | ○ | ○ | ○ | ○ | |
| | | | | During | Long | | ○ | ○ | ○ | ○ | |
| Johnston[47] 2 high intensity | iRCT | Unclear | Not reported | During | Long | | ○ | | ○ | | Two of four comparisons showed improved language outcomes, and one of three comparisons showed poorer SEWB outcomes, in the intervention arm in the long term, while intervention was ongoing. |
| Landry[48] 2 high intensity | iRCT | Unclear | 91% of parents completed all 10 visits plus 2 assessment visits | Post | Short | | ● | ● | ○ | | Two of two comparisons showed improved language outcomes, and one of one comparison showed improved cognitive outcome in the intervention arm. Three of six comparisons showed improved SEWB outcomes in the intervention arm. Interaction with birth weight examined, but reporting of results incomplete. |

Outcome measurement (1) Timing: During = while intervention is ongoing; Post = after intervention is completed; (2) When?: Immediate = <1 month; Short = 1–6 months; Medium = >6–12 months; Long = >12 months.
● = outcome reported, statistically significant differences in favour of intervention found in 70% or more of comparisons within a study.
Ø = outcome reported, statistically significant differences in favour of control found in 70% or more of comparisons.
○ = outcome reported, no statistically significant differences found or found in <70% of comparisons.
◇ = outcome reported, inconsistent results (defined as in table 3).
(blank box), outcome not reported.
Size of the symbol indicates the total sample size included in analysis: ●Ø○◇=≥500; ●○◇=100–500; •ø○=<100.
CGS, community groups; Cogn, cognitive; cRCT, cluster randomised controlled trial; Immed, immediate; iRCT, individually randomised controlled trial; IUGR, intra-uterine growth retardation; Lang, language; SEWB, social and emotional well-being; SHV, supportive health visiting.

in research of this kind. We had also hoped that this review would advance our knowledge on the types of social and emotional well-being outcomes that can be influenced by interventions of this kind. However, this was not possible given that the outcomes included were not well defined or consistent and mainly measured behaviour. Future studies that aim to measure effects on social and emotional well-being in young children need better articulation of their conceptual definitions of the social-emotional domains targeted[68] and the proposed mechanisms of action of the intervention. Finally, 15 studies did not publish a protocol or provide evidence of trial registration, and 2 did not report on all outcomes described in the Methods section of the paper. Improvements in trial registration and a priori specification of analysis plans are needed in trials in this field.

We also note that adherence was poor across studies and inconsistently reported. Future research should carefully report uptake, adherence and fidelity (particularly whether parents have received the intervention in sufficient dose) to further develop our understanding of the mechanism of action of these programmes and how to engage and retain families.[69 70] Involvement of parents from the design stage onwards is essential to improve engagement of families within these important research studies.[71] Recent work has shown that monetary incentives can also increase participant retention in RCTs.[72] Research is also needed on whether new delivery platforms (such as technology-assisted interventions[73]) may provide a more engaging, feasible and cost-effective mechanism for providing support to parents.

There have been calls for new public health models of interventions to enhance early child development within existing healthcare systems.[74] As shown in our review, however, the current evidence base for interventions delivered to all families is lacking. It is unclear from the literature reviewed why programmes had limited impact on child developmental outcomes. However, many of the interventions relied on parents to change their behaviours and action in relation to their children and were educational in tone but did not have a theoretical framework or a sound basis in behaviour change mechanisms.[75] Additionally, authors did not always report on a clear formative research phase or logic model. Future studies should follow guidance on the development and evaluation of complex interventions (such as the Medical Research Council's guidance).[76] The results of all phases of intervention development also need to be published alongside trial results, as current studies alone do not allow us to fully understand why interventions have not produced expected effects.

Currently, there is insufficient evidence that, where health services are available to all families with very young children, additional elements or enhancements to these improve child development outcomes. Early intervention to improve child development is a public health priority, but funding is scarce. There is an urgent need for more robust evaluation of existing interventions and to develop and evaluate novel intervention packages to enhance the offer to all families.

**Contributors** LH, SP, PL, DW, JW and RL conceived and designed the study. MM designed the search strategy (in consultation with other review authors) and performed the database searches. LH and LJG searched the websites and journal table of contents. LH and LJG selected and reviewed eligible reports. LH, LJG, SG, SP and TP extracted data. LH drafted the paper. All authors commented on and revised the paper, and approved the final version. LH is the guarantor for the paper.

**Funding** This work was funded by Public Health Wales. The Director of Policy, Research and Development at Public Health Wales (Professor MAB) provided expert technical advice during discussions on the study design, the interpretation of the results and the drafting of the paper, and is an author on the paper.

**Competing interests** None declared.

**Patient consent** Not required.

**Provenance and peer review** Not commissioned; externally peer reviewed.

**Data sharing statement** This paper reports on a systematic review. No original study data was obtained from the authors of the trials included in the review. Extra data (such as the results of the review stratified by characteristics other than those included in the report) is available by emailing Lisa Hurt.

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
