## [Reviewer comments · BMJ Open]

ARTICLE DETAILS

TITLE (PROVISIONAL)	Interventions that enhance health services for parents and infants to improve child development and social and emotional wellbeing in high-income countries: A systematic review
AUTHORS	Hurt, Lisa; Paranjothy, Shantini; Lucas, Patricia; Watson, Debbie; Mann, Mala; Griffiths, Lucy; Ginja, Samuel; Paljarvi, Tapio; Williams, Jo; Bellis, Mark; Lingam, Raghu

VERSION 1 – REVIEW

REVIEWER	David R Jones University of Leicester, UK
REVIEW RETURNED	09-Jan-2017

GENERAL COMMENTS	This is a pertinent and helpful review of a generally rather weak evidence base in the target area. The searching process appears to have been comprehensive. Presentation of methods and results is mostly clear (but see comments below). Cautious interpretation of results and the recommendations for further research to augment the results presented in this review are appropriately indicated. The summary of the results could be improved somewhat: 1) Presentation of the main results as +ve/no/ -ve effect is defective. Presentation of estimates and SDs in web appendix C is better, but explicit inclusion of 'best estimates' and confidence intervals from the better quality primary studies would be preferable. 2) Although the authors can justify not attempting meta-analysis, graphical presentation in the form of augmented forest plot summaries of key results, eg from the best studies, indicating outcome measure and other characteristics of importance, would be valuable. See NICE methods manual for public health guidelines PMG4 (Sept 2012) Fig 5.1 in section 5.4.4 (available at https://www.nice.org.uk/process/pmg4/chapter/reviewing-the-scientific-evidence).
---

REVIEWER	Shelly-Anne Li University of Toronto Toronto, Ontario Canada
REVIEW RETURNED	20-Jan-2017

GENERAL COMMENTS	Introduction: Page 4 line 52: "Embedding interventions within a service, such as health visiting, which provides universal, ongoing and consistent support for parents, may also improve the interaction between health professionals..." What do you mean by health visiting providing universal support for these parents? Health visiting may also be part of targeted home visiting interventions. Please be revise the writing here. Since your SR is targeted to high-income countries, I am surprised there was little information pertaining to high-income countries. Why did you choose to study this population as opposed to include low-income countries? Perhaps a line or two addressing this issue would help orient readers better. Methods: The list of databases searched was comprehensive and relevant to the research topic. The search strategy was focused and sensitive. However, it would have been helpful to include more synonyms that would capture outcomes more. Why did you search RCTs from 1996 to 2016? Inclusion/exclusion: What is the rationale of including two broad categories of studies? (i) compared interventions to enhance health professional contact in the very early years with usual care, or ii) compared different interventions with each other. Readers would benefit from a justification for this approach. Because this may promote heterogeneity in the review findings Page 6, Line 53: We were primarily interested in studies that used validated tools to measure these outcomes, but also included descriptive indicators of child development as secondary outcomes. What do you mean by 'primarily interested in'? Please communicate clearly whether you included studies that did not use validated tools to measure the outcomes. Study selection: Were the title, abstract and full text reviews conducted independently and in duplicate? Please specify if in duplicate. And if not in duplicate, why not? Also, when you stated two reviewers, the initials were provided. Please revise. Limitations: Please expand on the limitations of your study. For example, was the search limited to English language? Title: I'd recommend that the title of your review to reflect your evidence synthesis of universal health services. But further down in the results tables, you also included 'geographically targeted' interventions? If there are two meanings of 'universal' (one which you described in the introduction, and the other displayed in the results), please clearly define. Otherwise, please justify your inclusion of geographically targeted interventions, because from
--

	your objectives and introduction, it appears to focus on universal interventions. Analysis: What do you classify as 'low', 'moderate', or 'high intensity'? You stated that the review authors used subjective determination. Are there any standardized parameters these assessors were using? I find it interesting that there was high heterogeneity even with studies that investigated the same domain and used the same questionnaires (i.e. motor development). Please elaborate on how you arrived at the conclusion of the level of heterogeneity. For example, which parts of PICO (Participant, Intervention, Comparison, Outcomes) caused heterogeneity? So was it statistical heterogeneity or clinical heterogeneity? Or both? This review would have high benefited from meta-analyses. General comments: There appears to be too many outcomes in the original research question, which may have resulted in high heterogeneity. A major weakness of this review is, due to the large number of outcomes, this precluded review authors from exploring/communicating each of these domains in depth as they rightly deserve. But this review serves as a good overview of what the literature is reflecting about universal interventions.
--	--

REVIEWER	Rebecca PILLAI RIDDELL York University
REVIEW RETURNED	03-Feb-2017

GENERAL COMMENTS	Article Summary 1. Lines could be streamlined for summary but simply summarizing the main findings and 1 line of implications. Substantial methodological comments seem out of place. Introduction 1. Age reporting (page 5, line 21) Abstract refers to 3 years of age. Also in PROSPERO registration (CRD42015015468) refers to 3 months to 3 years but also up to 2 years. Then there is confluct between introduction and methods re: what is written lines 3-4 on page 7 (mean age 39 months or less) and page 6, lines 5-7 (mean age at 24 months). Please clarify throughout document. Methods 1. Having conducted at least a dozen published reviews in the past 2 years (including a large Cochrane), I had to say that the Tables tracking information are impressive and not typical in published systematic reviews outside the Cochrane. It reflects gold standard practice. It is clear for a reviewer to track why decisions were made and to evaluate syntheses. This is particularly important as at times subjective decisions were made instead of an a priori algorithm (e.g. 37-38 page 8)- a decision with which I agree. 2. One thought I had was in terms of publication of the core manuscript (assuming all 82 pages would not be included). I would argue Table 1-3 are likely most critical with web access to the rest of the materials. Could the key info from Table 4 be integrated in earlier
---

	tables. To me, I most wanted to know when looking at the first 3 Tables was which of the 5 domains they were tapping into (motor, socio-emotional, etc). Would have liked a simple column for that and less detail on study explanation. Details of which exact measures were used are not necessary as core reading (because of the extreme heterogeneity) but helpful to be able to refer to as an appendix or supplemental materials perspective. Results:  1. In the syntheses where 2 active interventions are compared, on a number of occasions the reviewers note “no effect”. I am not clear on what this means. A comparison of two active interventions with a non-significant effect size (SMD) would mean no difference not no effect. They could both be effective but not have a difference in magnitude of effect. Did the authors examine past the numbers to determine any within group comparisons were conducted within each group (i.e. pre- and post-intervention as opposed to between group)? This would seem the more appropriate data to synthesize for this review (e.g. lines 36-39 page11) 2. Subgroup effects: It was unclear if commentary was based on examining within a study (i.e. the study authors had to examine these effects) or broadly speaking over studies (The review authors examined this across studies). Out of the 22 studies, how many were in high versus low SES communities? Did this matter on positive outcomes? Discussion:  1. I think authors have a unique opportunity to comment on clear recommendations for the field . What should researchers should be doing. Were there measures that should be used across, were there components within trials that may have been effective or may suggest that further study is warranted. Any exemplar trials that could be held up? Common mistakes among trials- commentary about the 100% of trials that were not properly blinded (a key challenge in psychosocial trials but outcome assessment could absolutely be blinded). These are important issues that take the careful and methodical work of the team and can impact research in the field in order to improve medical care. Given results feel this is what the bulk of the discussion should centre on... 2. Why mention parent stakeholders in methods if no mention in discussion. Given parents not involved in study itself, feel mention should be omitted earlier.
--	--

REVIEWER	SERGIO FACCHINI PEDIATRIC PRIMARY CARE UNIT AAS 5, PORDENONE NATIONAL HEALTH SYSTEM ITALY
REVIEW RETURNED	20-Feb-2017

GENERAL COMMENTS	I really enjoyed reviewing this work. I deeply appreciated the author’s effort to offer a review on such a critically important topic: how to enhance the quality of the early year universal health services provision in high-income countries to allow all children reaching their full developmental potential.
---

Recent evidence support a life course perspective on childhood development through advances in neuroscience and longitudinal follow-up approaches. These findings provide strong economic justification for investment in early childhood, especially in children younger than three years. Preventive and promotive packages can build on existing delivery platform across sectors for providing parenting and child services at scale.

The recent Lancet series “Advancing early childhood development: from science to scale” (Lancet 2017; 389: 77-118) provided a comprehensive update analysis of universal early childhood interventions in low and middle-income countries building on existing services (ie health). High quality evidence of effectiveness in enhancing universal health professional contact strategies to improve early child development in high-income countries are lacking instead. The object of this review is therefore justified and needed!

This systematic review is built on a clearly defined protocol that describes in detail the object, rationale, hypothesis and planned method for the review and was an essential component of the review process quality. It enables the reader to rule out deviations from planned methods and bias (ie selective reporting of outcomes) in the interpretation of results and conclusions.

The adopted PRISMA reporting guideline enables the assessment of the completeness and transparency of the review protocol used. The authors may comment on why they used PRISMA 2009 guidelines and checklist for their systematic review protocol instead of PRISMA for Protocols 2015. I must say that even checking with PRISMA-P the protocol is robust and fulfils all the criteria.

As the protocol was registered, please provide the name of the registry (ie PROSPERO) and registration number in the abstract as required by PRISMA-P 2015 checklist.

The review clearly describe how the strength of the body of evidence was assessed. GRADE is a well-developed formal process to rate the quality of scientific evidence that offers a transparent and structured process for developing and presenting evidence summaries. The choice of GRADE is appropriate, as it is “outcome centric”. Another advantage of GRADE is that it required the systematic reviewer to make explicit his or her judgment of each factor that determines the quality of evidence for each outcome and then use the data from the individual eligible studies to generate a best estimate of the effect on each patient-important outcome and an index of the uncertainty associated with that estimate.

For each comparison all outcome are presented together in one evidence table (evidence profile). As this systematic review addressed more the one comparison more then one summary of findings table is presented. Another merit of this systematic review is that all relevant outcome are adequately addressed in a single document.

I’m not sure if it is possible to prepare a visual representation of the relationships between studies and outcomes (comparisons) but it could further improve the comprehension of the strength of the body of evidence.

Please notice that Web Appendix C (full narrative summary of results) is not mentioned in the text: I suppose that on line 21 at page 11 should be written Appendix C . While there is not a Web Appendix E (page 13 line 53): I suppose it is Appendix D.

As it was not possible to assume that studies were estimating the same treatment effect due to the heterogeneity of the studied population, intervention design and outcome measurement statistical meta-analysis of effectiveness data was not carried out. For this

	reason, a narrative approach to synthesizing the findings was undertaken instead of a quantitative approach. This option was not scheduled in the final review protocol. The methods of narrative synthesis process should ideally be specified in advance although this may not be always possible. I think that the process of narrative synthesis might be made more transparent by describing which of the tools and techniques were used and why in a specific concise section. A brief explanation of choices made for the development of the preliminary synthesis (ie tabulation, grouping, vote counting) and the exploration of relationships within and between studies (ie sub groups and moderator variable) would be appreciated. If any narrative synthesis guidance (ie Popay et al 2006. Guidance on the conduct of narrative synthesis in systematic reviews) was adopted to avoid bias it should be declared. It is not clear if two reviewers worked independently and then compared their findings to produce a mutually agreed final version. In summary, although the manuscript is methodologically sound, I feel that it might benefit from a brief additional critical reflection on the synthesis process making clear the choice of the tools and techniques used. It would be useful to have a brief general comment on why the interventions did not show any significant outcome despite all the evidence that such intervention should work. Apart from RCTs bias analysis, is there any other aspect of the studies analysed (ie theory of change, intervention type, context, health provider) on which authors can reflect upon? Eventually it would be nice to hear some more implications for future research (ie type of intervention, type of health provider, etc.).
--	---

REVIEWER	Symone Detmar TNO, Child Health, The Netherlands
REVIEW RETURNED	06-Mar-2017

GENERAL COMMENTS	This is a generally well-written review that examines the effect of interventions designed to enhance universal health services provided to parents and children in the first two years of life. It reviews a large amount of research. Despite the considerable effort and expertise this paper represents, there are several problematic issues in this review. 1. First, the rationale for examining such a broadly framed question (effectiveness of universal interventions) is not clear nor theoretically elaborated in the introduction. This review focuses on the effectiveness of universal interventions that promote maternal and child health during early years. "There is high-quality global evidence to support the effectiveness of such universal services, but most of these studies have been targeted on high risk families with an identified condition."(page 4). Studies with high-risk samples have been excluded from this review. However, there were eight studies included in the review with a targeted, more specific sample (page 9, line 34). This is confusing. What is exactly the research question in the review: the effectiveness of universal interventions in high-risk and low-risk populations? There is no literature reviewed on the differential effects of parental interventions in these subgroups. No clear hypotheses or expectations were formulated. For example: do the reviewers expect lower outcomes in low-risk populations? Neither is defined what is exactly meant by an
---

'universal intervention'. To conclude: the research question is very general and seems to be insufficiently informed by knowledge from previous studies.

2. I do not understand the remark at page 8 that "parents were not involved in the design and conduct of the review, but we are discussing the results and interpretation with parents". What is the rationale for the decision not to assess the effectiveness of the interventions on parental outcomes such as attitude, parental sensitivity of interaction style? All interventions that were investigated here, were primarily delivered to parents, and focuses directly on the parent, and indirectly on child outcomes. It seems most logical also to include the parental side of the picture. However, the decision not to focus on parents is not explained, nor justified on empirical or theoretical considerations.

3. This review does not seem to contribute significantly to our knowledge about parental interventions in the early years. On page 5 the authors mentioned: "Previous reviews of early interventions fail to provide a full picture of interventions to public health policy and practice because they do not provide a comprehensive examination of child development outcomes in the early years." (page 5, line 9-12). This is a firm statement. The objective of this review is to provide such a comprehensive picture. I seriously doubt whether this review can meet this expectations. The main outcome of the review can be summarized as: 1) the set of studies is very heterogeneous – which is rather obvious given the broad research question, 2) quality of evidence was generally low, 3) and the effectiveness of the interventions was inconsistent. What does these outcomes contribute to our knowledge about parental interventions in the early years? The results of this review do not appear to inform the reader of how processes work like improving parental skills, or improving parent-child relations or the extent to which interventions may be operating in particular circumstances or in subgroups.

4. The only remarkable outcome in this review is the finding that studies of 'high intensity of intervention' did not confer more benefit for the child than those of lower intensity (page 15). This finding is consistent with an earlier meta-analysis (Bakermans et al, 2003), but any explanation of this finding in the Discussion section lacks.

How the reviewer exactly came to the conclusion that higher intensity studies were no more effective, is not described in a convincing way. In the review, three levels of study intensity and three levels of risk of bias are distinguished, resulting in different small subgroups of studies (Table 4). On page 13 the outcomes of different subgroups were described, for example: "In the studies at low risk of bias, there was no intervention effect when either low or high intensity interventions were studied. Some positive effects were seen in the two-trials of moderate intensity interventions, although in one, this was limited to subgroups only (children with disturbed attachment at baseline, and in the other positive effects were not consistently seen." (page 13, line 26 – 33). One weaker study showed inconsistent positive effects in one subgroup. And remaining studies that provided to less information about the risk factors, neither revealed clear outcomes. The overall picture the reader gets here is that "there was no clear pattern in the results" (line 44 and line 51). Why should the inconsistent picture justify the conclusion that high intensity interventions were no more effective than low intensity interventions?

5. The methodology of this review is rather weak or flawed. I mention below some more specific issues:

- Inclusion of the studies:

The authors identified a large amount of 15.000 articles but somehow ended up with only 22 articles or trials for the review. Procedures for identifying studies and exclusion criteria need greater specification. The Figure 1 PRISMA flow diagram is helpful to understand the selection process, but the biggest step (from 12,986 to 342 studies) remains rather unclear. The only information that is provided is that 12,644 records “were outside the scope of the review” (page 9, line 10). The Methods section mentioned three inclusion (RCT, embedded in routine healthcare, and universal intervention) and two exclusion criteria (age of the child at the start of the intervention younger than 2 years, and no high-risk groups). Can these five criteria justify the exclusion of about 12.500 studies? Additionally, the exclusion process itself is a bit confusing: from the total set of 22 trials in this review, there were even eight studies (about 30%) with a ‘targeted’ sample: which comes very close to ‘a high risk sample’. The more fundamental problem here is that this review may be dependent on a vague or subjective decision that a large amount of trials were ‘outside the scope of the review’.

- Procedure of vote counting is flawed:

The authors did much effort to code different sources of bias, following Cochrane guidelines. This procedure ends up in different subgroups of studies, which can be very informative in meta-analysis if effect sizes in different subgroups can statistically be compared to each other. However, “it was not possible to conduct a meta-analysis due to the heterogeneity in the types of interventions and methods used to measure outcomes” (page 14, line 37, 38). A bit strange to mention such an important consideration in the Discussion section. Appendix C contains all information that was extracted from the primary studies (also means and standard deviations), but these findings cannot be integrated or summed up into a combined effect size. The reviewer seems to rely more heavily on the verbal information provided in the primary study (effect, no effect). Thus, it seems to me that simple frequency counts (known as ‘vote counting’) were used to summarize the study outcomes: how many primary studies reported a positive effect, how many reported ‘no effect’ and how many studies reported ‘mixed effects’. This vote-counting procedure is a weak strategy according to Cochrane guidelines. The narrative reviewer can search patterns in the outcomes, but the significance of findings in primary studies depends on the sample size. A narrative review cannot take into account this dependency, whereas meta-analysis can. Thus, despite the enormous and careful work this article represents, I do not think that this review really adds a significant contribution to the field. Why not focusing on a more precise research question? I would advise to reshape/rework this manuscript into a meta-analysis, or a set of meta-analyses on more specific hypotheses. Meta-analysis enables the reviewer to compute combined effect sizes for subgroups of studies. Such a meta-analysis can really add new information to the knowledge about parental interventions, for example by finding evidence for small but relevant effects that would have remain unknown in primary studies.

refs

Bakermans-Kranenburg, M. J., Van IJzendoorn, M. H., & Juffer, F.

	(2003). Less is more: meta-analyses of sensitivity and attachment interventions in early childhood. Psychological Bulletin , 129, 195-215.
--	---

VERSION 1 – AUTHOR RESPONSE

Reviewer 1

Comment

Response

4

1) Presentation of the main results as +ve/no/ -ve effect is defective. Presentation of estimates and SDs in web appendix C is better, but explicit inclusion of 'best estimates' and confidence intervals from the better quality primary studies would be preferable.

In response to this comment, we have:

1. Changed the wording in Table 3 (Summary of Findings), replacing “positive effect” with either “better outcomes in intervention than control” or “better outcomes in the more intensive intervention group”, and replacing negative effect with “worse outcomes in intervention than control” or “worse outcomes in the more intensive intervention group” (highlighted in yellow in the table);

2. Replaced Table 4 with an effect direction plot (see response to comment 5 for more detail);

3. Edited the sentence signposting the table in the Web Appendix which provides effect estimates as follows (page 13, line 357):

“A full narrative summary of the results, including the tools used to assess the outcome in each trial and the estimates of intervention effects, is given in Web Appendix C.”;

4. Added the effect estimates from the studies at low risk of bias to the text summary of each outcome (see pages 13-16).

5

2) Although the authors can justify not attempting meta-analysis, graphical presentation in the form of augmented forest plot summaries of key results, eg from the best studies, indicating outcome measure and other characteristics of importance, would be valuable.

Producing augmented forest plots for these results is not possible due to the lack of consistency in outcome measures used. However, we have investigated alternative options for producing a visual display of the results as suggested by this reviewer and reviewer 4. As a result, we have replaced Table 4 with an effect direction plot (Thomson and Thomas 2012). The purpose of these is to provide a “visual summary of complex

quantitative data to accompany more detailed tables and narrative synthesis”. We adapted the design used by Hartwell et al (2016) for our plot. We now refer to this plot on page 13 (line 363), page 16 (lines 454 and 459), and page 17 (line 474).

Thomson H, Thomas S. The effect direction plot: visual display of non-standardised effects across multiple outcome domains. *Res Syn Meth* 2013;4:95–101.

Hartwell G, et al. E-cigarettes and equity: a systematic review of differences in awareness and use between sociodemographic groups. *Tob Control* Published Online First: 21 December 2016. doi: 10.1136/tobaccocontrol-2016-053222

Reviewer 2

Comment

Response

6

Page 4 line 52: "Embedding interventions within a service, such as health visiting, which provides universal, ongoing and consistent support for parents, may also improve the interaction between health professionals..." What do you mean by health visiting providing universal support for these parents? Health visiting may also be part of targeted home visiting interventions. Please be revise the writing here.

See response to comment 2 above.

We have removed the word "universal" from the specific sentence mentioned by this reviewer (page 5, line 122).

7

Since your SR is targeted to high-income countries, I am surprised there was little information pertaining to high-income countries. Why did you choose to study this population as opposed to include low-income countries? Perhaps a line or two addressing this issue would help orient readers better.

An overview of interventions to promote early child development in low and middle income countries has recently been published (Britto et al 2017, as part of the Lancet Advancing Early Childhood Development series). A reference to this paper has been added in the introduction (page 6, line 125 onwards) and the following sentence edited to read:

"However, the effectiveness of such interventions to enhance existing multi-disciplinary services in high income settings is not known." (page 6, line 129)

Britto P, Lye S, Proulx K, et al. Nurturing care: promoting early childhood development. *Lancet* 2017;389:91-102.

8

The list of databases searched was comprehensive and relevant to the research topic. The search strategy was focused and sensitive. However, it would have been helpful to include more synonyms that would capture

Electronic databases search retrieved over 15,000 records. Nevertheless, to maximise recall and minimise the potential for publication bias, a range of supplementary techniques was used to increase the sensitivity of the search and to ensure coverage of grey literature and unpublished studies. These included searching websites of 58

outcomes more.

relevant programmes and organisations, electronic table of contents of eight key journals, checking reference lists of included papers, authors contacted for additional data and reviewing systematic reviews on relevant topics for primary studies meeting inclusion/exclusion criteria. Three of the 22 included studies have not yet been published in peer-reviewed journals. We therefore believe that our search was comprehensive.

9

Why did you search RCTs from 1996 to 2016?

We were interested in interventions that were relevant for inclusion in services as they are delivered now. We therefore felt that including studies published over the past 20 years would provide the most pertinent evidence.

We have added a sentence providing this justification, which reads:

“Given our focus on enhancement of existing health services, we restricted to studies published within 20 years of our study inception since health service change has been substantial in the mid to late 20th century.” (page 8, line 216)

10

What is the rationale of including two broad categories of studies? (i) compared interventions to enhance health professional contact in the very early years with usual care, or ii) compared different interventions with each other. Readers would benefit from a justification for this approach. Because this may promote heterogeneity in the review findings

We agree that it is confusing to make a statement about this in the section on inclusion and exclusion criteria, as we did not decide on this categorisation a priori; this distinction became apparent when extracting the data from the included studies. We have therefore removed this from page 6, line 152. As recommended for a Cochrane Summary of Findings table, we have maintained the distinction in the presentation of the results.

In the section where we describe the interventions, we have added the following sentence to clarify the aim of the studies where two interventions were compared:

“In the five studies that included two interventions, the interventions were of the same intensity in all but one (Doyle, which compared a medium intensity intervention with one of high intensity), and they aimed to compare different models of care with each other.” (page 12, line 343)

We do not believe that these different comparisons led to heterogeneity in the review findings. We might have expected to see more positive intervention effects in studies comparing an intervention to usual care than in the studies comparing two different interventions, but this was not the case. We have added a sentence stating this into the first paragraph of the discussion:

“Studies that compared one intervention with usual care did not demonstrate more positive intervention effects than studies comparing two interventions.” (page 17, line

494)

11

Page 6, Line 53: We were primarily interested in studies that used validated tools to measure these outcomes, but also included descriptive indicators of child development as secondary outcomes.

What do you mean by 'primarily interested in'? Please communicate clearly whether you included studies that did not use validated tools to measure the outcomes.

We agree that this sentence is unclear and have changed it as follows:

“We included studies that used validated tools to measure these outcomes. Where unvalidated tools were used, we considered these to be secondary outcomes.” (page 8, line 203)

We have now also included the outcome domains studied in each trial in Table 1 (as suggested by reviewer 3), and have indicated in this table whether the outcomes were measured using validated tools in each trial.

12

Were the title, abstract and full text reviews conducted independently and in duplicate? Please specify if in duplicate. And if not in duplicate, why not? Also, when you stated two reviewers, three initials were provided. Please revise.

We have amended this text to read:

“Titles and abstracts were screened for inclusion independently by two of three reviewers (LH and LJG or SP). Full text versions were obtained for the papers potentially meeting the inclusion criteria and were screened independently by two of three reviewers (LH and LJG or SP).” (page 9, line 233)

13

Limitations: Please expand on the limitations of your study. For example, was the search limited to English language?

This is stated on page 7, line 161 and now additionally we state:

“We included randomised controlled trials (with individual or cluster randomisation), in any language, that were published or unpublished.” (page 6, line 151)

14

Title: I'd recommend that the title of your review to reflect your evidence synthesis of universal health services.

The title has been amended to:

“Interventions that enhance health service contact with parents and infants to improve child development and social and emotional wellbeing in high-income countries: A systematic review”

15

But further down in the results tables, you also included 'geographically targeted' interventions? If there are two meanings of 'universal' (one which you described in the introduction, and the other displayed in the results), please clearly define. Otherwise, please justify your inclusion of geographically targeted interventions, because from your objectives and introduction, it appears to focus on universal interventions.

See our response to comment 2.

To clarify that this is what we mean by “geographic targeting”, we have edited the sentence which describes this to read:

“Studies that selected participants from the general population or included all individuals from a specific neighbourhood (for example, an area-based programme defined on the basis of postcode or zip code, known as “geographically targeted” programmes in this review) were included.” (page 7, line 172)

There is also a description of how many studies did this, and examples of how they defined the geographic area on page 12 (line 317), and we have not changed this sentence.

16

What do you classify as 'low', 'moderate', or 'high intensity'? You stated that the review authors used subjective determination. Are there any standardized parameters these assessors were using?

The seven criteria that we used to classify the intensity of the intervention are given on page 10, line 266. Previous Cochrane reviews (for example, Baker et al 2015) have used a similar process, to allow for the diversity and complexity of different interventions. We note that reviewer 3 states that the method we used was appropriate (see comment 21).

We have edited the sentence in which we include this reference to clarify this (see page 10, line 272). We also provide examples of the interventions classified in each of the three groups in the results section (see page 12, line 329 onwards).

Baker P, Francis D, Soares J, et al. Community wide interventions for increasing physical activity. Cochrane Database of Systematic Reviews 2015;Issue 1:Art. No.: CD008366.

17

I find it interesting that there was high heterogeneity even with studies that investigated the same domain and used the same questionnaires (i.e. motor development). Please elaborate on how you arrived at the conclusion of the level of heterogeneity. For example, which parts of PICO (Participant, Intervention, Comparison, Outcomes) caused heterogeneity? So was it statistical heterogeneity or clinical heterogeneity? Or both? This review would have high benefited from meta-analyses.

We state (on page 9, line 249) that “due to heterogeneity in the i) the populations studied, ii) the design of the interventions and iii) the wide range of outcome measures used (both in terms of the child development domains and/or the instruments used to assess the outcomes)”.

We agree that our use of the word “heterogeneity” in this context is confusing, as it may imply statistical or clinical heterogeneity. We have changed this word to “variation” throughout the document.

We also agree that it would have been helpful to conduct a meta-analysis, had the included studies been amenable to this. To clarify that a meta-analysis was not possible, we have also edited the above sentence to read:

“Due to variation in i) the populations studied, ii) the design of the interventions and iii) the wide range of outcome measures used (both in terms of the child development domains and/or the instruments used to assess the outcomes), it was not possible to conduct a meta-analysis and results were reported using narrative synthesis.”

18

There appears to be too many outcomes in the original research question, which may have resulted in high heterogeneity. A major weakness of this review is, due to the large number of outcomes, this precluded review authors from exploring/communicating each of these domains in depth as they rightly deserve.

But this review

We thank the reviewer for noting that our review provides a good overview of these interventions.

As noted in our response to comment 3, our focus was intended to be consistent with the ethos and design of early years services that view the child as a whole, and expect the workforce within those services to promote child development overall. We also note that,

serves as a good overview of what the literature is reflecting about universal interventions.

given the paucity of studies, we believe that multiple small reviews would each have little to contribute, so the challenge of reporting many is offset by the ability to provide a comprehensive summary across many.

Reviewer 3

Comment

Response

19

Lines could be streamlined for summary but simply summarizing the main findings and 1 line of implications. Substantial methodological comments seem out of place.

We have done this (see page 4, lines 70 onwards).

20

Age reporting (page 5, line 21) Abstract refers to 3 years of age. Also in PROSPERO registration (CRD42015015468) refers to 3 months to 3 years but also up to 2 years. Then there is conflict between introduction and methods re: what is written lines 3-4 on page 7 (mean age 39 months or less) and page 6, lines 5-7 (mean age at 24 months). Please clarify throughout document.

We examined interventions delivered from the antenatal period to two years postpartum. As some programmes continued beyond the child's second birthday, we included studies where the mean age at inclusion was 24 months or less. To allow time for these interventions to produce demonstrable effects, we examined outcomes to three years postpartum (39 months or less, given that not all studies would manage to assess children on their 3rd birthday exactly).

To clarify this, we have:

1. Reported these ages in months throughout (either 24 months or 39 months);
2. Added a sentence in the section on inclusion and exclusion criteria which reads:

“To allow time for these interventions to produce demonstrable effects, we included studies that examined outcomes to 39 months of age (given that not all studies would manage to assess children on their 3rd birthday exactly).” (page 7, line 167)

21

Having conducted at least a dozen published reviews in the past 2 years (including a large Cochrane), I had to say that the Tables tracking information are impressive and not typical in published systematic reviews outside the Cochrane. It reflects gold standard practice.

It is clear for a reviewer to track why decisions were made and to evaluate syntheses. This is particularly important as at times subjective decisions were made instead of an a priori algorithm (e.g. 37-38 page 8)- a decision with which I agree.

We thank the reviewer for their comments.

22

One thought I had was in terms of publication of the core manuscript (assuming all 82 pages would not be included). I would argue Table 1-3 are likely most critical with web access to the rest of the materials. Could the key info from Table 4 be integrated in earlier tables. To me, I most wanted to know when looking at the first 3 tables was which of the 5 domains they were tapping into (motor, socio-emotional, etc). Would have liked a simple column for that and less detail on study explanation.

Details of which exact measures were used are not necessary as core reading (because of the extreme heterogeneity) but helpful to be able to refer to as an appendix or supplemental materials perspective.

We have edited Table 4 as described above (this now contains an effect direction plot).

This has allowed us to edit the other tables, as requested by this reviewer, as follows:

1. Table 1 now includes a column on the outcome domains measured in each study. We have also included information in the footnote describing which studies used validated tools to assess the outcomes;
2. The information on who conducted the intervention has been moved to Table 2, to sit alongside the other information on the characteristics of the intervention;
3. The adherence information is included in Table 4 along with the effect direction plot.

As noted in our response to comment 4, we have also edited the sentence to provide clearer signposting to the supplemental information on outcome measurement and estimates of intervention effects (see page 13, line 357 onwards).

23

In the syntheses where 2 active interventions are compared, on a number of occasions the reviewers note "no effect". I am not clear on what this means. A comparison of two active interventions with a non-significant effect size (SMD) would mean no difference not no effect. They could both be effective but not have a difference in magnitude of effect.

Did the authors examine past the numbers to determine any within group comparisons were conducted within each group (i.e. pre- and post-intervention as opposed to between group)? This would seem the more appropriate data to synthesize for this review (e.g. lines 36-39 page11) ...

We agree that "no difference" is more appropriate terminology when two interventions were compared with each other, and have changed this throughout the results section (lines 386, 401, 421, and 446) and in Table 3 (changes highlighted in yellow).

The reviewer also makes a good point about the need to clarify how the outcomes were compared when the trial included two interventions. In all five trials that compared two interventions, the comparison presented was the difference in the outcome measures at the endpoint of the trial between the two trial arms (not the difference in change between the two outcomes). One of the studies that compared an intervention with usual care did examine the difference in the change (Niccolls 2008).

To clarify this, we have included a footnote in Table 3 that states that:

“The comparison reported is the ratio or difference in estimates between intervention and control group, or between the two intervention groups, at the specified follow-up point unless otherwise noted (** indicates where the difference in change between intervention and control is used instead).”

24

Subgroup effects: It was unclear if commentary was based on examining within a study (i.e. the study authors had to examine these effects) or broadly speaking over studies (The review authors examined this across studies). Out of the 22 studies, how many were in high versus low SES communities? Did this matter on positive outcomes?

We examined sub-group effects within a study where the authors presented these, and across studies with different population characteristics.

No conclusions can be drawn on sub-group effects examined in the individual studies (for example, the socio-economic status of participants) because of sparse data and incomplete reporting of the analyses in the individual papers (see page 16, line 452). We have changed the sub-heading for this section to “Sub-group effects reported within studies”.

We also examined the results of the studies, stratified by characteristics of the study populations (for example, were the interventions delivered in a specific geographic location or not?, tables included in web appendix D). As stated on page 17 (line 480), no clear patterns to suggest an effect in studies including certain populations (for example, high versus low SES communities) was seen, although the details presented on the characteristics of the recruited participants (and whether these were representative of the target populations of the studies) were limited in most papers. We have changed the sub-heading for this section to “Stratification of results across studies by risk of bias and intensity of interventions”.

25

I think authors have a unique opportunity to comment on clear recommendations for the field. What should researchers should be doing. Were there measures that should be used across, were there components within trials that may have been effective or may suggest that further study is warranted. Any exemplar trials that could be held up? Common mistakes among trials- commentary about the 100% of trials that were not properly blinded (a key challenge in psychosocial trials but outcome assessment could absolutely be blinded). These are important issues that take the careful and methodical work of the team and can impact research in the field in order to improve medical care. Given results feel this is what the bulk of the discussion should centre on...

We agree with the reviewer that we should provide additional detail in the discussion on this.

To this end, we have:

1. Edited the sentence at the end of the first paragraph of the discussion and added more detail about the methodological issues that were consistent across the trials. This now reads:
“The low-to-moderate quality of evidence overall suggests that there is a need for high quality robust trials to inform current health service delivery in this area.” (page 17, line 498)
2. Added a paragraph on the methodological weaknesses of the trials, and made recommendations for improvements in future studies (page 19, line 544).

3. Added a paragraph in the discussion on the lack of detail in these publications on the theoretical frameworks or logic models used, and how these need to be published alongside trial results before we can fully understand the findings and replicate the

implementation of interventions such as these (page 20, line 570).

4. Added suggestions on the need for increased parent involvement in the design of programmes and research (page 19, line 563), and the possible use of incentives to improve participant retention (page 19, line 565).

26

Why mention parent stakeholders in methods if no mention in discussion. Given parents not involved in study itself, feel mention should be omitted earlier.

A statement on any public involvement included in all studies is a requirement for BMJ journals, so we retain for this reason. We believe it is important to acknowledge this lack in our own research.

We were not sure whether the reviewer was also commenting here about the need to discuss parental outcomes in the discussion. See our response to comment 40 below where we address this.

Reviewer 4

Comment

Response

27

As the protocol was registered, please provide the name of the registry (ie PROSPERO) and registration number in the abstract as required by PRISMA-P 2015 checklist.

This has been added at the end of the abstract.

28

I'm not sure if it is possible to prepare a visual representation of the relationships between studies and outcomes (comparisons) but it could further improve the comprehension of the strength of the body of evidence.

See response to comment 5 above.

Table 4 now contains an effect direction plot, which we hope provides this visual representation.

29

Please notice that Web Appendix C (full narrative summary of results) is not mentioned in the text: I suppose that on line 21 at page 11 should be written Appendix C . While there is not a Web Appendix E (page 13 line 53): I suppose it is Appendix D.

We have corrected these typographic errors (see page 13, line 359 and page 17, line 481).

30

I think that the process of narrative synthesis might be made more transparent by describing which of the tools and techniques were used and why in a specific concise section.

A brief explanation of choices made for the development of the preliminary synthesis (ie tabulation, grouping, vote counting) and the exploration of relationships within and between studies (ie sub groups)

All study screening, data extraction, classification of the intensity of the intervention, the risk of bias, and the GRADE quality of evidence was conducted independently by 2 reviewers. Both reviewers also used the same data extraction sheet.

We did not specifically follow Popay's 2006 guidance, although the methods we used were consistent with their guidance. For example, the textual descriptions of each study was produced using a systematic format, including the same information for all studies

and moderator variable) would be appreciated. If any narrative synthesis guidance (ie Popay et al 2006. Guidance on the conduct of narrative synthesis in systematic reviews) was adopted to avoid bias it should be declared. It is not clear if two reviewers worked independently and then compared their findings to produce a mutually agreed final version.

and in the same order. Findings were compared and the few discrepancies noted were resolved by the 2 reviewers where possible. Where this was not possible, we discussed with other team members and reached a consensus.

In addition, the results presented in the summary of findings table do represent a form of vote counting (the tabulation of statistically significant and non-significant findings). We felt that this was important information to include because several of the studies included a large number of statistical comparisons but found few statistically significant effects (which are unduly emphasised in their reporting of the results). This leads to the most important conclusion as stated at the end of the first paragraph of the conclusion:

“the low-to-moderate quality of the evidence suggests that there is a need for high quality robust trials to inform current health service delivery in this area” (page 17, line 498 onwards)

31

It would be useful to have a brief general comment on why the interventions did not show any significant outcome despite all the evidence that such intervention should work. Apart from RCTs bias analysis, is there any other aspect of the studies analysed (ie theory of change, intervention type, context, health provider) on which authors can reflect upon?

Most of these interventions tried to educate parents in some way about child development, and promote activities that they could undertake to improve their child's developmental outcomes. Most described the body of literature on which the intervention had been based but did not describe the development-feasibility-evaluation cycle set out in the MRC guidance on the development of complex interventions. There is therefore limited information available to help us conclude why the interventions have not worked.

To reflect this uncertainty, we have added a sentence on the theoretical basis for the interventions in the description of the interventions that reads:

“Most papers described the body of literature on which the intervention development had been based, but provided less detail on the proposed mechanisms of action of the intervention.” (page 12, line 327)

We have also added sentences in the discussion that read:

“Future studies should follow guidance on the development and evaluation of complex interventions (such as the Medical Research Council's guidance).

The results of all phases of intervention development also need to be published alongside trial results, as current studies alone do not allow us to fully understand why interventions have not produced expected effects.” (page 20, line 577)

32

Eventually it would be nice to hear some more implications for future research (ie type of intervention, type of health provider, etc.).

See our response to comment 25 where we detail how we have expanded on the implications for future research in the discussion.

Reviewer 5

Comment

Response

33

This is a generally well-written review that examines the effect of interventions designed to enhance universal health services provided to parents and children in the first two years of life. It reviews a large amount of research.

We thank the reviewer for their comment.

34

First, the rationale for examining such a broadly framed question (effectiveness of universal interventions) is not clear nor theoretically elaborated in the introduction... To conclude: the research question is very general and seems to be insufficiently informed by knowledge from previous studies.

See our response to comment 3, which describes the changes we have made to clarify the purpose of the review and the justification for our approach.

35

“There is high-quality global evidence to support the effectiveness of such universal services, but most of these studies have been targeted on high-risk families

with an identified condition.”(page 4). Studies with high-risk samples have been excluded from this review. However, there were eight studies included in the review with a targeted, more specific sample (page 9, line 34). This is confusing.

See our response to comment 15, where we clarify what is meant by “geographic targeting”.

These are not targeted programmes in the same way as programmes designed for children with specific conditions or families at high-risk of adverse outcomes, as all individuals within a defined population are eligible for inclusion.

For example, one of the “geographically targeted” programmes was delivered in Camden and Islington – two London boroughs with high levels of both wealth and deprivation. The programme was open to women from “deprived enumeration districts” in these boroughs but all women living in those districts were eligible for inclusion, regardless of their individual wealth or any other characteristics.

What is exactly the research question in the review: the effectiveness of universal interventions in high-risk and low-risk populations? There is no literature reviewed on the differential effects of parental interventions in these subgroups.

The aim of the review is to examine the evidence for interventions that enhance health service contacts delivered to all families. We hope that the changes that we have made in response to comment 2 clarify this in the paper.

We specified a priori that we would examine the results stratified by whether the programme was available to all or “geographically targeted” (as specified on page 10 line 256). This stratification is included in Web Appendix D, and we state on page 17 (line 480) that “no clear pattern in the results were seen when stratification by the other pre-specified variables was conducted (see Web Appendix D)”.

No clear hypotheses or expectations were formulated. For example: do the reviewers expect lower outcomes in low-risk populations?

We agree that we did not specify our hypotheses for the examination of the results stratified by certain trial or population characteristics. We have added a sentence giving our justification for selecting these variables on page 10 (line 257):

We selected these variables as we hypothesised that they would help to identify the characteristics of the interventions most likely to be effective (for example, if high intensity interventions were more effective than low intensity ones) or the populations in which they were most likely to be effective (for example, if programmes recruiting from certain neighbourhoods were more effective than those made available to all).

Neither is defined what is exactly meant by an ‘universal intervention’.

See our response to comment 3, where we acknowledge the difficulty with the definition of “universal” and describe how we have edited the text to clarify this.

I do not understand the remark at page 8 that “parents were not involved in the design and conduct of the review, but we are discussing the results and interpretation with parents”.

See our response to comment 25.

A statement describing any public involvement in a project is a requirement for BMJ journals.

What is the rationale for the decision not to assess the effectiveness of the interventions on parental outcomes such as attitude, parental sensitivity of interaction style? All interventions that were investigated here, were primarily delivered to parents, and focuses directly on the parent, and indirectly on child outcomes. It seems most logical also to include the parental side of the picture. However, the decision not to focus on parents is not explained, nor justified on empirical or theoretical considerations.

The stated aim of these interventions was to improve child development outcomes. They hoped to achieve this by educating or changing the behaviour of parents, which in turn would lead to improvements in developmental outcomes.

Whilst we agree that it is important to understand whether the parents had increased their knowledge or changed their behaviour, we also suggest that this provides information on the process by which the intervention would be expected to improve child development outcomes, rather than being the primary aim of the interventions.

We agree that this should be stated explicitly in the paper, and we have edited the sentence in the discussion where we mention parental outcomes to read:

“Previous studies have examined the effects of programmes such as these on parental knowledge, attitudes or practices. We did not systematically review parental outcomes here, so cannot comment on whether parents benefitted from these interventions. However, we can conclude that – in these studies – any effects on the

parents did not, in turn, lead to consistent improvements in child development outcomes.” (page 18 line 521)

41

This review does not seem to contribute significantly to our knowledge about parental interventions in the early years. On page 5 the authors mentioned: “Previous reviews of early interventions fail to provide a full picture of interventions to public health policy and practice because they do not provide a comprehensive examination of child development outcomes in the early years.” (page 5, line 9-12).

This is a firm statement. The objective of this review is to provide such a comprehensive picture. I seriously doubt whether this review can meet this expectation.

We acknowledge the concerns of this reviewer. However, we also note that Reviewers 1, 2 and 4 describe the review as “pertinent and helpful”, “meaningful”, and covering a “critically important topic”.

The sentence referred to by the reviewer states that previous reviews do not provide a comprehensive picture of interventions “...in the very early years” (that is, the first 24 months of life).

We believe that we have summarized all of the available evidence for interventions delivered alongside health service contacts to all families during this time period in this review.

42

The main outcome of the review can be summarized as: 1) the set of studies is very heterogeneous – which is rather obvious given the broad research question, 2) quality of evidence was generally low, 3) and the effectiveness of the interventions was inconsistent. What does these outcomes contribute to our knowledge about parental interventions in the early years?

There have been calls for new public health models of interventions to enhance early child development within existing healthcare systems (for example, the Healthy Child Programme in England). The groups who plan these want to embed evidence-based programmes within their services. The contribution of this review, therefore, is to state that there are interventions that have been developed for this purpose, but that the evidence for their effectiveness is currently weak, both because of the methodological weaknesses of the studies that examine them and because they fail to articulate their theoretical frameworks or their logic models.

We hope, as specified in our response to comment 25, that the additions we have made to the discussion section now make this clear.

The results of this review do not appear to inform the reader of how processes work like improving parental skills, or improving parent-child relations or the extent to which interventions may be operating in particular circumstances or in subgroups.

We agree that this is a weakness in the reporting of the trials themselves. In response to this (and comment 25), we have added the following text in the discussion section:

“It is unclear from the literature reviewed why programmes had limited impact on child developmental outcomes. However, many of the interventions relied on parents to change their behaviours and action in relation to their children and were educational in tone, but did not have a theoretical framework or a sound basis in behaviour change mechanisms. Additionally, authors did not always report on a clear formative research phase or logic model. Future studies should follow guidance on the development and evaluation of complex interventions (such as the Medical Research Council’s guidance) The results of all phases of intervention development also need to be published alongside trial results, as current studies alone do not allow us to fully understand why interventions have not produced expected effects.” (page 20 line 572)

The only remarkable outcome in this review is the finding that studies of ‘high intensity of intervention’ did not confer more benefit for the child than those of lower intensity (page 15). This finding is consistent with an earlier meta-analysis (Bakermans et al, 2003), but any explanation of this finding in the Discussion section lacks. How the reviewer exactly came to the conclusion that higher intensity studies were no more effective, is not described in a convincing way. ... Why should the inconsistent picture justify the conclusion that high intensity interventions were no more effective than low intensity interventions?

We agree that we could elaborate on the lack of an evident dose-response relationship in these studies. We have therefore extended the sentence in the discussion where we mention this, to read:

“There was also no evidence that interventions of high intensity confer more benefit than those of lower intensity as no dose-response relationship was evident: programmes of greater intensity (in terms of length, number or type of components) did not show more positive intervention effects than programmes of lower intensity.” (page 18 line 536)

The methodology of this review is rather weak or flawed.

We worked with policy makers from Public Health England and Wales to ensure that our research question was of direct relevance to their work, worked with an information specialist to ensure that our searches were comprehensive, and followed Cochrane Library and PRISMA guidance in the conduct of the review.

We note that reviewer 3 states that our methods “reflect gold standard practice” (see comment 21).

The authors identified a large amount of 15.000 articles but somehow ended up with only 22 articles or trials for the review. Procedures for identifying studies and exclusion criteria need greater specification. The Figure 1 PRISMA flow diagram is helpful to understand the selection process, but

the biggest step (from 12,986 to 342 studies) remains rather unclear. The only information that is provided is that 12,644 records “were outside the scope of the review” (page 9, line 10). The Methods section mentioned three inclusion (RCT, embedded in routine healthcare, and universal

We agree that it would be helpful to give some additional detail on the articles that we excluded from the review. We have therefore added the text in parentheses to the results section:

“After title and abstract screening, 12,644 records that were outside the scope of the review were excluded (the vast majority of these because their intervention was targeted at families at high-risk of adverse outcomes or at children with identified conditions).” (page 11, line 288)

intervention) and two exclusion criteria (age of the child at the start of the intervention younger than 2 years, and no high-risk groups). Can these five criteria justify the exclusion of about 12.500 studies?

47

However, “it was not possible to conduct a meta-analysis due to the heterogeneity in the types of interventions and methods used to measure outcomes” (page 14, line 37, 38). A bit strange to mention such an important consideration in the Discussion section.

See our response to comment 16, which explains that we have added text in the Methods section stating that a meta-analysis was not possible.

48

The reviewer seems to rely more heavily on the verbal information provided in the primary study (effect, no effect). Thus, it seems to me that simple frequency counts (known as ‘vote counting’) were used to summarize the study outcomes: how many primary studies reported a positive effect, how many reported ‘no effect’ and how many studies reported ‘mixed effects’. This vote-counting procedure is a weak strategy according to Cochrane guidelines. The narrative reviewer can search patterns in the outcomes, but the significance of findings in primary studies depends on the sample size. A narrative review cannot take into account this dependency, whereas meta-analysis can.

To draw conclusions on the results from each study, we examined the data presented in each study, as summarised in the table in Web Appendix C. We agree that, in trying to condense these for the “Summary of Findings” table, we over-simplified the results.

See our response to comment 4, where we describe how we have added further details of the results into the body of the paper.

See also our response to comment 30 on the issue of vote counting. Several of the studies conducted a large number of statistical tests and we felt that it was important for readers to be able to appreciate that only a small minority of these tests led to statistically significant results.

49

Thus, despite the enormous and careful work this article represents, I do not think that this review really adds a significant contribution to the field. Why not focusing on a more precise research question? would advise to reshape/rework this manuscript into a meta-analysis, or a set of meta-analyses on more specific hypotheses. Meta-analysis enables the reviewer to compute combined effect sizes for subgroups of studies.

See our response to comment 3 where we explain why we do not believe that this review should be re-shaped into several reviews, and the changes we have made to strengthen the justification for our approach.

We agree that it would have been helpful to conduct a meta-analysis, had the included studies been amenable to this. See our response to comment 16, which details the changes we have made to the paper to clarify why a meta-analysis was not possible.

VERSION 2 – REVIEW

REVIEWER	David (R) Jones University of Leicester, UK
REVIEW RETURNED	02-May-2017

GENERAL COMMENTS	I am a little disappointed that more effort was not made to present key results graphically.
--

REVIEWER	Shelly-Anne Li University of Toronto
REVIEW RETURNED	17-May-2017

GENERAL COMMENTS	Although I do appreciate the broad search strategy, I'd suggest reviewers to develop a clear and defined research question for this review. The present manuscript provides little contribution to advancing our knowledge about which types of social and emotional well-being outcomes are associated with interventions that enhance health service contact with parents and infants. For instance, I'd be very interested in learning more about which types of social and emotional well-being outcomes are being frequently investigated, and what were these studies' conclusions? I feel that the reviewers are mainly focused on assessing the quality of the study methods (risk of bias, how these studies were reported), instead of synthesizing the outcomes. I highly recommend for reviewers to revisit their research question, and be very explicit about what the review is trying to accomplish (both in the abstract and introduction, as well as in the limitations section).
--

REVIEWER	SERGIO FACCHINI PEDIATRIC PRIMARY CARE UNIT AZIENDA PER L'ASSISTENZA SANITARIA N° 5 FVG NATIONAL HEALTH SYSTEM ITALY
REVIEW RETURNED	08-May-2017

GENERAL COMMENTS	I'm satisfied that all requested changes have been made. Let me add some further brief comments on this considerable work. The investigation of the effectiveness of interventions to promote early child development in high-income countries is needed. The review research question makes sense when you consider its purpose to help policy makers and providers to enhance existing health services with effective implementation programs. The focus on the broad field of child development (and its domains) is therefore appropriate as this is often the focus of universal intervention in the early years. I agree that this review should not be reshaped in many different little reviews. The visual display of key results with an effect direction plot (table 4) is satisfying. The new added paragraphs with comments on recommendations for future research add valuable insight to new projects planning for the field. Of notice is the suggestion to follow the MRC guidance on the development of complex intervention with explicit description of the mechanism of action of the intervention (how parents change their behaviour and how this would affect their children). This type of interventions are quite complex to study and implement. While ECD universal interventions through health services are effective in low middle-income countries (ECD Lancet series 2017) it is much harder to find the key element to add for further enhance the rich environment in which most high-income children live in. Even more some interventions can work in one context but not in another even in high-income countries: the recent paper on the effectiveness of adding the intensive home-visiting intervention NFP (Robling et al Lancet 2016; 387:146-55) to the usually provided health and social care provided in England showed so far no additional benefit to the selected primary outcomes. Even if it is disappointing to find that the evidence for effective universal interventions to enhance child development in existing high-income countries health services is lacking, I believe that this review could make providers more cautious to implement any existing program and finance more sound research on the topic.
---

VERSION 2 – AUTHOR RESPONSE

I am a little disappointed that more effort was not made to present key results graphically.

We explored different methods of presenting the results graphically but, given the differences in the outcome measures and populations included, none were appropriate except for the effect direction plots.

For example, in the six studies that examined motor development outcomes:

- Two presented mean motor development scores from the Bayley Scales of Infant Development (measured at 4 months of age in one study, and at 12 months of age in the other study);
- One presented mean motor development scores from the Development Profile II at 24 months of age;
- One presented the mean hand/eye coordination score from the Griffith Mental Development scale
 - One presented data for gross and fine motor skills separately (from the Ages and Stages questionnaire at various ages up to 3 years);

- One used video-coded scores for the supine, prone, sitting, and standing sub-scales of the Alberta Infant Motor Scale, as well as the mean age at which particular motor milestones had been reached.

Summarising these results graphically (for example, using augmented Forest plots) would not be possible. We also explored the other potential graphical presentation methods described in Popay's 2006 guide for conducting narrative synthesis (including Harvest plots). The most appropriate alternative for our data was the effect direction plot that we have now included in the paper.

We note that reviewer 4, who also requested additional graphical representation of the results, commented that "the visual display of key results with an effect direction plot (table 4) is satisfying."

We have not therefore made any further additions to the paper in response to this comment.

Reviewer 2

Comment

Response

4

Although I do appreciate the broad search strategy, I'd suggest reviewers to develop a clear and defined research question for this review... I highly recommend for reviewers to revisit their research question, and be very explicit about what the review is trying to accomplish (both in the abstract and introduction, as well as in the limitations section).

We acknowledge the reviewer's comments and concede that our question is broad. However, we would also argue that this was unavoidable due to the nature of the topic. The research question was developed by public health researchers in collaboration with public health practitioners and an information specialist. The question was developed in response to an identified need for high-quality evidence to support the implementation of interventions within routine health services if they were found to lead to improvements in child development outcomes.

We used similar methods to those used by other researchers in the field, such as a recent rapid review of interventions for consideration for the Healthy Child Programme in the UK (Axford N, Barlow J, Coad J, et al. Rapid Review to Update Evidence for the Healthy Child Programme 0–5. London, UK: Public Health England, 2015). We believe that the fact that we found 22 trials, the majority of which examined interventions with a broad remit and examined a wide range of child development outcomes, justifies our approach.

We also note that reviewer 4's comments contradict the comments of this reviewer:

"The investigation of the effectiveness of interventions to promote early child development in high-income countries is needed. The review research question makes sense when you consider its purpose to help policy makers and providers to enhance existing health services with effective implementation programs. The focus on the broad

field of child development (and its domains) is therefore appropriate as this is often the focus of universal intervention in the early years... Even if it is disappointing to find that the evidence for effective universal interventions to enhance child development in existing high-income countries health services is lacking, I believe that this review could make providers more cautious to implement any existing program and finance more sound research on the topic."

We have not therefore made any changes to the paper based on these comments.

5

The present manuscript provides little contribution to advancing our knowledge about which types of social and emotional well-being outcomes are associated with interventions that enhance health service contact with parents and infants. For instance, I'd be very interested in learning more about which types of social and emotional well-being outcomes are being frequently investigated, and what were these studies' conclusions?

We agree that it is disappointing that we could not draw firmer and clearer conclusions on the social and emotional wellbeing outcomes in this review. We now acknowledge this in the discussion (page 18, line 525) and included a new reference that could be used by future studies to improve the conceptualisation of social and emotional wellbeing outcomes. The new text reads:

“We had also hoped that this review would advance our knowledge on the types of social and emotional well-being outcomes that can be influenced by interventions of this kind. However, this was not possible given that the outcomes included were not well-defined or consistent, and mainly measured behaviour. Future studies that aim to measure effects on social and emotional wellbeing in young children need better articulation of their conceptual definitions of the social-emotional domains targeted⁶⁸ and the proposed mechanisms of action of the intervention.”

New reference (number 68): Denham S, Wyatt T, Bassett H, et al. Assessing social-emotional development in children from a longitudinal perspective. *J Epidemiol Community Health* 2009;63(Suppl 1):i37–i52.

We have also clarified in the presentation of the results that the social and emotional well-being outcomes measured in these trials were mainly behavioural outcomes (page 13, line 392), and we have provided better signposting to Table C4 of the web appendices on page 13, line 388 (where information on all of the social and emotional well-being outcomes and the conclusions of the studies that examined these can be found).

6

I feel that the reviewers are mainly focused on assessing the quality of the study methods (risk of bias, how these studies were reported), instead of synthesizing the

We have presented the information of the risk of bias within and across studies in accordance with PRISMA guidelines. Popay (2006) also argues that “if primary studies of poor methodological quality are included in the review in an uncritical manner then this

outcomes.

will affect the trustworthiness of the synthesis”. In this review, the most striking finding is that no intervention effect was seen in any of the high-quality trials, reinforcing our key message that there is a need for high-quality robust trials to inform current health service delivery in this area.

We have not therefore made any changes to the paper based on this comment.

Reviewer 4

Comment

Response

5

I'm satisfied that all requested changes have been made.

We thank the reviewer for their detailed and positive comments, and their acknowledgement that we have made all requested changes.

VERSION 3 – REVIEW

REVIEWER	David R Jones University of Leicester, UK
REVIEW RETURNED	23-Jul-2017

GENERAL COMMENTS	The effort made to present key results graphically has now been demonstrated!
---

REVIEWER	Shelly-Anne Li University of Toronto, Canada
REVIEW RETURNED	18-Aug-2017

GENERAL COMMENTS	All reviews have been addressed, and suggested changes are revised.
---

REVIEWER	SERGIO FACCHINI Pediatric Primary Care Unit National Health System AAS 5 Friuli Occidentale Pordenone, Italy
REVIEW RETURNED	30-Jul-2017

GENERAL COMMENTS	I appreciate the effort to improve further this systematic review. Probably some minor modifications are still possible but I believe that this edition targets efficaciously an important topic and rises even more important research questions in a detailed and clear way. Universal health services promote prevention as well as mental health provision. Advocates highlight the need of early intervention defined by mental health research and embedded in universal primary care health services. Falling short of these expectations by not being able to provide health services that attain optimal physical, mental and social health and well-being for all infants undermines the universal health system effectiveness. Many early prevention efforts yield clear value into adulthood (Campbell F, Conti J, Heckman JJ, et al. Early childhood investments substantially boost adult health. Science . 2014; 343: 1478-1485), but the precise
---

	mechanism through which these effects are obtained remain to be determined. I wish that this review could promote sound research in the field that will hopefully provide some clearly defined effective preventive strategy to promote early global (social and emotional in particular!) development in the universal health provision setting of high-income countries.
--	---